# Long-read sequencing unveils *IGH-DUX4* translocation into the silenced *IGH* allele in B-cell acute lymphoblastic leukemia

Liqing Tian [1], Ying Shao[1], Stephanie Nance[2], Jinjun Dang[2], Beisi Xu [1], Xiaotu Ma[1], Yongjin Li[1], Bensheng Ju[1], Li Dong[1], Scott Newman [1], Xin Zhou[1], Patrick Schreiner[1], Elizabeth Tseng[3], Ting Hon[3], Meredith Ashby[3], Chunliang Li[4], John Easton[1], Tanja A. Gruber[2] & Jinghui Zhang [1]

*IGH@* proto-oncogene translocation is a common oncogenic event in lymphoid lineage cancers such as B-ALL, lymphoma and multiple myeloma. Here, to investigate the interplay between *IGH@* proto-oncogene translocation and *IGH* allelic exclusion, we perform long-read whole-genome and transcriptome sequencing along with epigenetic and 3D genome profiling of Nalm6, an *IGH-DUX4* positive B-ALL cell line. We detect significant allelic imbalance on the wild-type over the *IGH-DUX4* haplotype in expression and epigenetic data, showing *IGH-DUX4* translocation occurs on the silenced *IGH* allele. In vitro, this reduces the oncogenic stress of *DUX4* high-level expression. Moreover, patient samples of *IGH-DUX4* B-ALL have similar expression profile and *IGH* breakpoints as Nalm6, suggesting a common mechanism to allow optimal dosage of non-toxic *DUX4* expression.

[1] Department of Computational Biology, St. Jude Children's Research Hospital, Memphis, TN 38105, USA. [2] Department of Oncology, St. Jude Children's Research Hospital, Memphis, TN 38105, USA. [3] Pacific Biosciences, Menlo Park, CA 94025, USA. [4] Department of Tumor Cell Biology, St. Jude Children's Research Hospital, Memphis, TN 38105, USA. Correspondence and requests for materials should be addressed to J.Z. (email: jinghui.zhang@stjude.org)

In B-cell development, the allelic exclusion of *IGH* (immunoglobulin heavy chain) follows a one B-cell, one antibody paradigm by first epigenetically silencing both alleles, then activating one allele during VDJ recombination so that functional *IGH* is expressed from only one active *IGH* allele in pre-B (precursor B) cells[1–4]. In lymphoid lineage cancer like B-cell acute lymphoblastic leukemia (B-ALL)[5], lymphoma[6], and multiple myeloma[7], *IGH@* proto-oncogene translocation is a common driver event. There is, however, limited knowledge about the interplay between *IGH@* oncogenic translocation and allelic exclusion of *IGH* (i.e., one allele is active, and the other is silenced) partly due to the technical challenges in delineating allelic specificity at the highly complex *IGH* locus (standard next-generation sequencing [NGS] technology can only generate short-read sequence data ranging from 100 to 150 bp).

*IGH-DUX4* translocation is one such oncogenic event, defining a B-ALL subtype with a distinct expression profile[8–10]. *DUX4* is a transcription factor located within the GC-rich D4Z4 repeat array at the subtelomeric regions of 4q35 and 10q26, which are characterized by high levels of repression[11,12]. It is expressed only during events associated with major chromatin relaxation, i.e., in early embryos at cleavage stage (2/4/8-cell embryos)[13] or after loss of repression of the D4Z4 macrosatellite repeat in myoblasts of individuals with facioscapulohumeral muscular dystrophy (FSHD)[14,15]. The *IGH-DUX4* subtype accounts for ~7% of pediatric B-ALLs in which an inter-chromosomal translocation repositions *DUX4* within the vicinity of the *IGH* enhancer (Eμ), the native enhancer of *IGH* gene (Igμ) in pre-B cell[8,9,16]. *IGH-DUX4* translocation is a clonal event acquired early in leukemogenesis of this B-ALL subtype, resulting in an aberrant activation of *DUX4* which is absent from other B-ALL subtypes[8,9].

In this study, we investigate the interplay between *IGH@* oncogenic translocation and allelic exclusion of *IGH* in Nalm6, a B-ALL cell line that harbors *IGH-DUX4* translocation. We use long-read technology which overcomes the limitation of short-read sequencing to evaluate the allele specificity of gene expression. Epigenetic states and enhancer–promoter interactions are analyzed using data generated from ChIP-seq, ATAC-seq, and 3-D genome assays. Significant allelic imbalance is detected on the wild type over the *IGH-DUX4* haplotype, showing that *IGH-DUX4* translocation occurs on the silenced *IGH* allele. In vitro assays suggest this may reduce the oncogenic stress of high-level expression of *DUX4*. Patient samples of *IGH-DUX4* B-ALL have similar expression profiles and *IGH* breakpoints as Nalm6, suggesting this could be a common mechanism; further analyses of haplotype structure and epigenetic profiling are required.

## Results

**Expression of *DUX4* and Igμ in *IGH-DUX4* B-ALLs**. Using published RNA-seq data generated from 32 B-ALL patient samples that harboring *IGH-DUX4* translocation and human embryos[9,13,17], we found that *DUX4* expression is much higher compared with normal expression in human cleavage stage embryos (the median FPKM [fragments per kilobase of transcript, per million mapped reads] 145.4 vs. 6.65, Fig. 1a). Typically this would suggest that the translocation would be into the active *IGH* allele. Intriguingly, however, *DUX4* expression was also much lower than Igμ in these B-ALL samples (the median FPKM 145.4 vs. 661.9, Fig. 1b)— raising an alternative possibility that *IGH-DUX4* translocation might have instead occurred on the silenced *IGH* allele.

**Selection of Nalm6 for genomic and epigenomic profiling**. Nalm6, an *IGH-DUX4* B-ALL cell line, was established from a 19-year-old patient with a near diploid B-ALL genome

(http://bioinformatics.hsanmartino.it/cldb/cl3632.html). Consistent with our observation in *IGH-DUX4* B-ALL patient samples, *DUX4* expression in Nalm6 is much higher compared with its normal expression in human cleavage stage embryos (FPKM 41.3 vs. 6.65, Fig. 1a) and much lower than Igμ (FPKM 41.3 vs. 638.2, Fig. 1b), making the cell line a good-model for evaluating allele specificity of the *IGH-DUX4* translocation.

To resolve the mapping ambiguity caused by high repeat content in *IGH* and *DUX4* regions, we profiled Nalm6 using three different approaches: (1) Iso-Seq, a method that generates full-length transcripts; (2) 10X Chromium whole-genome sequencing (WGS), which incorporates barcodes into large DNA fragments prior to WGS thereby producing the so-called "digital" long WGS reads; and (3) stranded total RNA-seq. We also performed epigenetic and chromosome conformation capture assays to evaluate the allele specificity of epigenetic states and enhancer–promoter interactions. These include (1) H3K27ac ChIP-seq, ATAC-seq, and whole-genome bisulfite sequencing (WGBS); (2) genome-wide 3-D genome assays including Hi-C and H3K27ac HiChIP; and (3) deep sequencing of targeted chromosome conformation capture (Capture-C).

**Allele specificity of *IGH-DUX4* translocation**. A previous report has shown that two *DUX4* copies were translocated between *IGHJ4* and *IGHD2–15* in Nalm6, resulting in expression of a *DUX4* protein with its last 16 amino acids replaced by *IGH*[8]. Incorporating this knowledge in our analysis, we first assembled the haplotype of *IGH-DUX4* translocation by selecting reads near the rearranged region using Chromium molecular barcodes. This resulted in a ~23 kb DNA sequence comprised of *IGHM*, the *IGH* intronic enhancer (Eμ), two copies of the translocated *DUX4* sequence, and flanking sequences near the two translocation breakpoints (Supplementary Fig. 1a). The translocated *DUX4* sequence shares 100% sequence identity to *DUX4L13* located on chromosome 10 while its best match on chromosome 4 is the *DUX4L4* locus with 99.67% (four mismatches) sequence identity. RNA-seq reads share 100% identity to the assembled *DUX4* contig, supporting exclusive transcription of *DUX4L13* activated by translocation from chromosome 10 to *IGH* in Nalm6 (Supplementary Fig. 2). The *IGH-DUX4* translocation in Nalm6 therefore occurred between chromosomes 10 and 14.

By mapping long Iso-Seq RNA reads to this assembled *IGH-DUX4* haplotype, we identified full-length transcripts representing functional Igμ, *IGH-DUX4* fusion, and antisense *DUX4* transcription, and all Iso-Seq RNA reads that contain *DUX4* are chimeric and can be mapped to the assembled *IGH-DUX4* haplotype (Supplementary Fig. 1b). The antisense *DUX4* transcript is comprised of an antisense *DUX4* segment joined to the *IGH* constant region (*IGHM*), which is also contained in the functional Igμ transcript. Bidirectional transcription of sense and antisense *DUX4* was also confirmed in our stranded RNA-seq data (Supplementary Fig. 2).

The *IGHM* locus is diploid in Nalm6 and contain three heterozygous germline SNPs (rs1059713, rs1136534, and rs3751511). Haplotype phasing from 10X Chromium WGS data demonstrate two haplotypes harboring alleles "T-G-G" and "A-A-A" of these three SNPs. The full-length transcripts generated by Iso-Seq showed that functional Igμ was transcribed exclusively from the "A-A-A" haplotype (i.e., the wild type without rearrangement) while the antisense *DUX4* was transcribed from the "T-G-G" haplotype that harbors the *IGH-DUX4* translocation (Fig. 2a). Therefore, *IGH-DUX4* translocation and functional Igμ are on different *IGH* alleles.

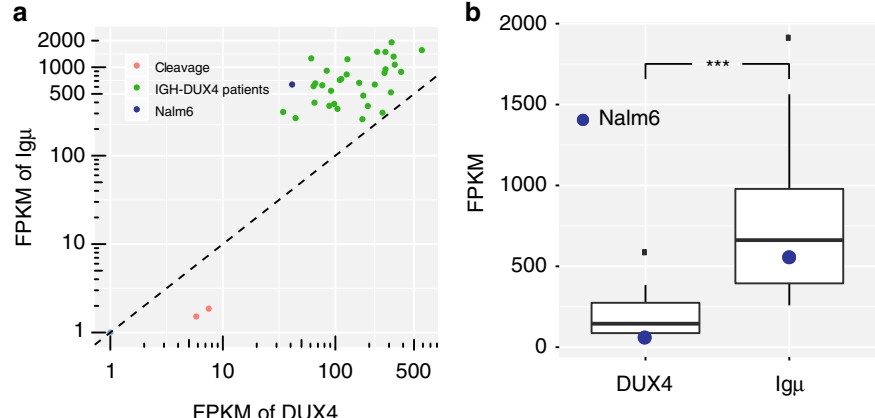

**Fig. 1** Expression of the translocated *DUX4* and Igµ in *IGH-DUX4* B-ALLs. **a** *DUX4* and Igµ expression in human embryonic cleavage cells, 32 *IGH-DUX4* B-ALL patients, and Nalm6 cell line are shown in red, green and blue, respectively. **b** Box plot of *DUX4* and Igµ expression level in the 32 *IGH-DUX4* B-ALL patients and Nalm6 cell line (blue dots). Median FPKM of *DUX4* and Igµ in 32 patients are 145.4 and 661.9, respectively. One-tailed, paired sample *t*-test was performed: ****p* < 0.001. Boxes show the first to third quartile with median, whiskers encompass 1.5 times of the interquartile range, and data beyond that threshold indicated as outliers

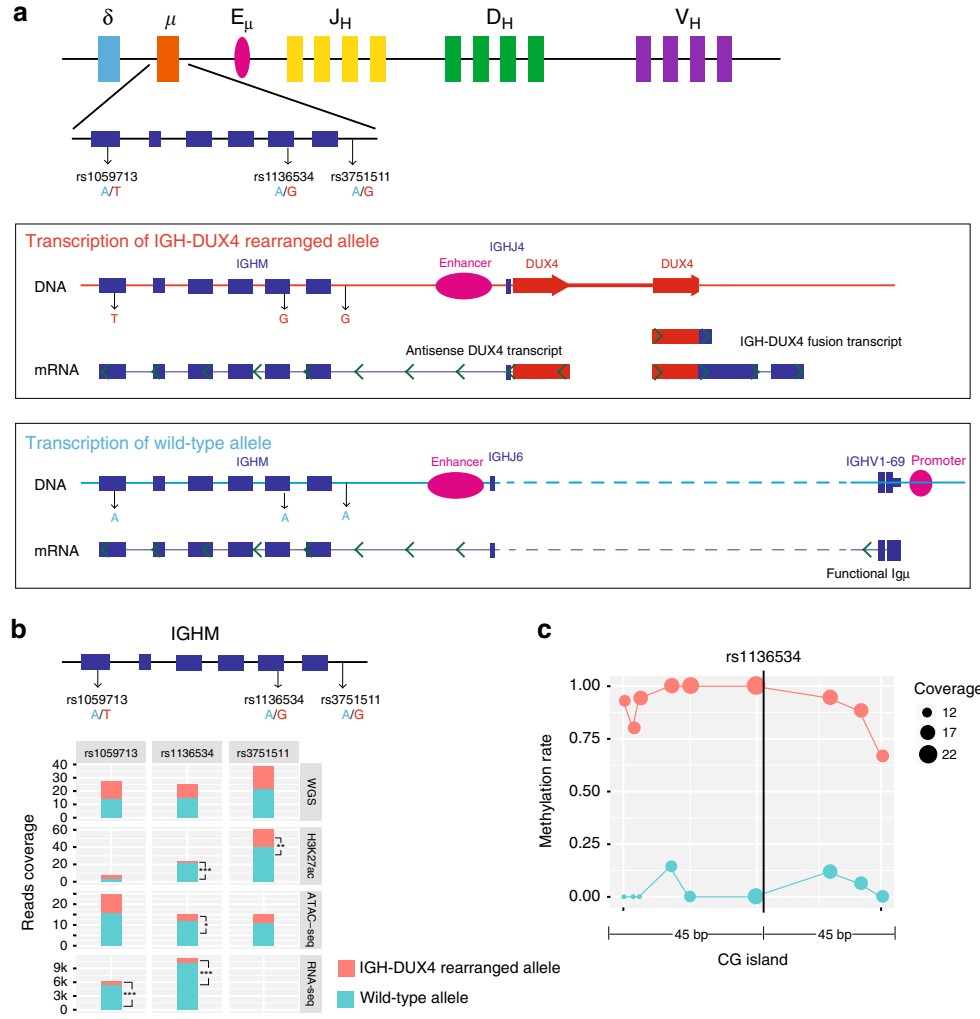

**Fig. 2** Allele specificity of *IGH-DUX4* translocation and functional Igµ. **a** Diagram of *DUX4* and *IGH* gene expression at two alleles in Nalm6. Functional Igµ and antisense *DUX4* are expressed from different alleles. Green arrows in transcripts indicate the direction of transcription. **b** Allelic imbalance of H3K27ac ChIP-seq, ATAC-seq, and RNA-seq data in the three heterozygous SNPs (rs1059713, rs1136534, rs3751511) located at *IGHM*. For RNA-seq, read-out is available only for the two exonic SNPs (rs1059713, rs1136534). One-tailed binomial test (expected probability 0.5) was performed: **p* < 0.05, ***p* < 0.01, ****p* < 0.001. **c** Evaluation of allelic imbalance of DNA methylation near the SNP rs1136534 located in a CG island region. Each CpG site near the SNP is shown by a circle drawn in proportion to the reads coverage used for calculating the methylation ratio (see Methods)

**IGH-DUX4 translocation on silenced IGH allele**. Prior studies have shown that functional Igμ is expressed only from the active IGH allele[1,3,18], suggesting that IGH-DUX4 translocation might be on the silenced IGH allele. Therefore, we examined allelic imbalance in RNA-seq, H3K27ac ChIP-seq, and ATAC-seq at the above-mentioned three heterozygous SNPs at the IGHM locus. Indeed, RNA-seq reads were significantly enriched on the wild-type IGH haplotype (i.e., the A-A-A haplotype) compared with the IGH-DUX4 haplotype (i.e., the T-G-G haplotype): expression of the wild-type allele is 7.4 and 10.4-fold of that of the IGH-DUX4 haplotype allele at the two exonic SNPs rs1059713 and rs1136534, respectively. A similar pattern was found in H3K27ac and ATAC-seq reads which measure enhancer activity and chromatin accessibility, respectively (Fig. 2b). One SNP (rs1136534) was also located in a CG island region, and DNA methylation data shows hypomethylation of the wild-type haplotype in contrast to hypermethylation of the IGH-DUX4 haplotype (Fig. 2c). These data provided further support that the DUX4 translocation occurs on the silenced IGH allele in Nalm6.

**Enhancer–promoter interaction at the two IGH haplotypes**. Prior studies have shown that both Igμ and the translocated DUX4 were regulated by the IGH enhancer (Eμ)[8,16]. We confirmed the interactions of Eμ-Igμ promoter and Eμ-DUX4 by performing genome-wide Hi-C (Fig. 3a) and H3K27ac HiChIP in Nalm6 (Supplementary Fig. 3). The Hi-C data demonstrate that the interaction of IGH locus on chromosome 14 with the DUX4 region on 10q26 is much broader than that with the DUX4 region on 4q35 (Supplementary Fig. 4), consistent with our finding that the translocated DUX4 was from chromosome 10 based on the assembled IGH-DUX4 haplotype and expressed transcripts by RNA-seq and Iso-Seq.

Eμ-Igμ promoter interaction measured by read-pair count in Hi-C and H3K27ac HiChIP is 3- and 8-fold of that of the Eμ-DUX4 interaction, respectively (Fig. 3c, details in Methods section). The stronger Eμ and Igμ interaction is not related to the distance-associated chromatin interaction decay[19] because the distance between Eμ and Igμ promoter on the wild-type haplotype (~140 kb) is in fact much longer than that (~4 kb) between Eμ and DUX4 on the IGH-DUX4 haplotype (Fig. 3b, see Methods). To verify this, we designed Capture-C with bait probes around Eμ and performed coverage-based peak analysis, which shows that Eμ-Igμ promoter interaction is 13-fold higher than that of Eμ-DUX4 interaction (Fig. 3c, see Methods). The uniform pattern of weaker Eμ-DUX4 interaction emerging from Hi-C, HiChIP, and Capture-C data provides further support that IGH-DUX4 haplotype was epigenetically silenced, consistent with allele-specific hypermethylation of this haplotype (Fig. 2c). The Eμ-Igμ promoter interaction was mono-allelic for the wild-type IGH allele based on the read-out of Hi-C/HiChIP/Capture-C data at the three heterozygous SNPs located near Igμ promoter (Supplementary Fig. 5b). This can be attributed to a reciprocal rearrangement between the 10q26 subtelomeric region and the IGH region unveiled by the Hi-C data (Supplementary Fig. 5, details in Methods section).

**DUX4 overexpression leads to increased apoptosis**. Our analysis on the transcriptome, epigenome and 3D genome shows that in Nalm6 the IGH-DUX4 translocation occurred on the silenced IGH allele, which would result in reduced expression of DUX4 compared with the alternative, i.e., translocation to the active, wild-type allele. Overexpression of oncogenes in cells may cause replicative stress and activation of anti-oncogenic pathways leading to apoptosis, a process often referred to as "oncogenic stress"[20,21]. Previous studies have demonstrated that transduced

expression of DUX4 in rhabdomyosarcoma cell lines led to apoptosis[22]. We therefore reasoned that the IGH-DUX4 translocation on the repressive haplotype would avoid high-level DUX4 expression if it were transcribed from the haplotype with an active Eμ which would not be tolerated in normal or cancer cells. To test this hypothesis, we introduced the DUX4 oncogene into hematopoietic cells by transduction of primary murine bone marrow cells with a retroviral vector that carries DUX4 and the eGFP reporter gene (Fig. 4a, b). Apoptosis, as measured by annexin V staining was determined in GFP high and low populations 36–48 h after transduction. We found a significant increase in apoptosis in cells with high levels of DUX4 compared with those with low levels as indicated by GFP levels ($p = 0.0484$). To further evaluate the effects of DUX4 overexpression in fully transformed leukemia cells, we introduced DUX4 into Nalm6 cells which already harbor the IGH-DUX4 translocation, confirming GFP-tagged DUX4 protein expression by western blot (Fig. 4c, d). The results showed that fully transformed leukemia cells would not tolerate further overexpression of DUX4 with a significant increase in apoptosis when comparing DUX4 transduced cells with the empty vector control ($p = 0.0124$). We therefore conclude that while low levels of the DUX4 protein are tolerable, high levels induce apoptosis providing a biologic rationale for expression from the repressive haplotype.

## Discussion

By integrating DNA translocation breakpoints, epigenetic profiling, transcription and cell toxicity data in Nalm6, we propose the following model for IGH-DUX4 translocation during B-cell development (Fig. 5) based on current knowledge of stepwise epigenetic process that controls the allelic exclusion of IGH[1,3,18]. B-cell development initiates from hematopoietic stem cells with both IGH alleles hypo-acetylated and silenced by DNA hypermethylation[1,18]. This is followed by the development of early progenitor B (pro-B) cells where both IGH alleles undergo D-to-J rearrangement. Structural variation (SV) breakpoints of IGH-DUX4 in Nalm6 occurred in the IGH D-J junction, suggesting that the IGH translocation arise at the pro-B cell stage as demonstrated previously by SV breakpoint analysis in IGH rearranged multiple myeloma patient samples[23]. Consistent with this, prior studies by our group and the others showed that the IGH-DUX4 fusions are clonal events acquired early in leukemogenesis[8,9]. At this stage, both IGH alleles remain methylated. In late pro-B and pre-B cell stage, one of the IGH alleles is randomly selected for activation by demethylation and hyperacetylation followed by the VDJ rearrangement. If an IGH-DUX4 rearranged allele were selected for activation—which is possible as not all B-ALLs express functional IGH[24,25], the resulting highly expressed DUX4 would likely be too toxic to permit the survival of leukemia cells, leading to cell death (Fig. 4). By contrast, activation of the wild-type IGH allele would lead to moderate expression of DUX4 from the silenced IGH-DUX4 allele, ensuring leukemia cell viability.

Previously, Yasuda et al. has shown that knockdown of the IGH-DUX4 fusion would suppress proliferation of Nalm6 cells[8], confirming the oncogenic potential of DUX4 expression in B-ALL. This, coupled with the oncogenic toxicity caused by overexpression of DUX4 presented in this study (Fig. 4), suggests that DUX4 expression in Nalm6 may follow the "Goldilocks principle"[26]—i.e., "just-right" levels are required as too-much will lead to apoptosis while too-little will lead to suppression of proliferation. Indeed, among patient samples that harbor IGH-DUX4 fusion, DUX4 expression represents a fraction of the Igμ expression level (26.8% +/−6.4% 95% CI) with a positive correlation (Pearson's $r = 0.554$, $p$-value = 0.001, two-tailed $p$ value by

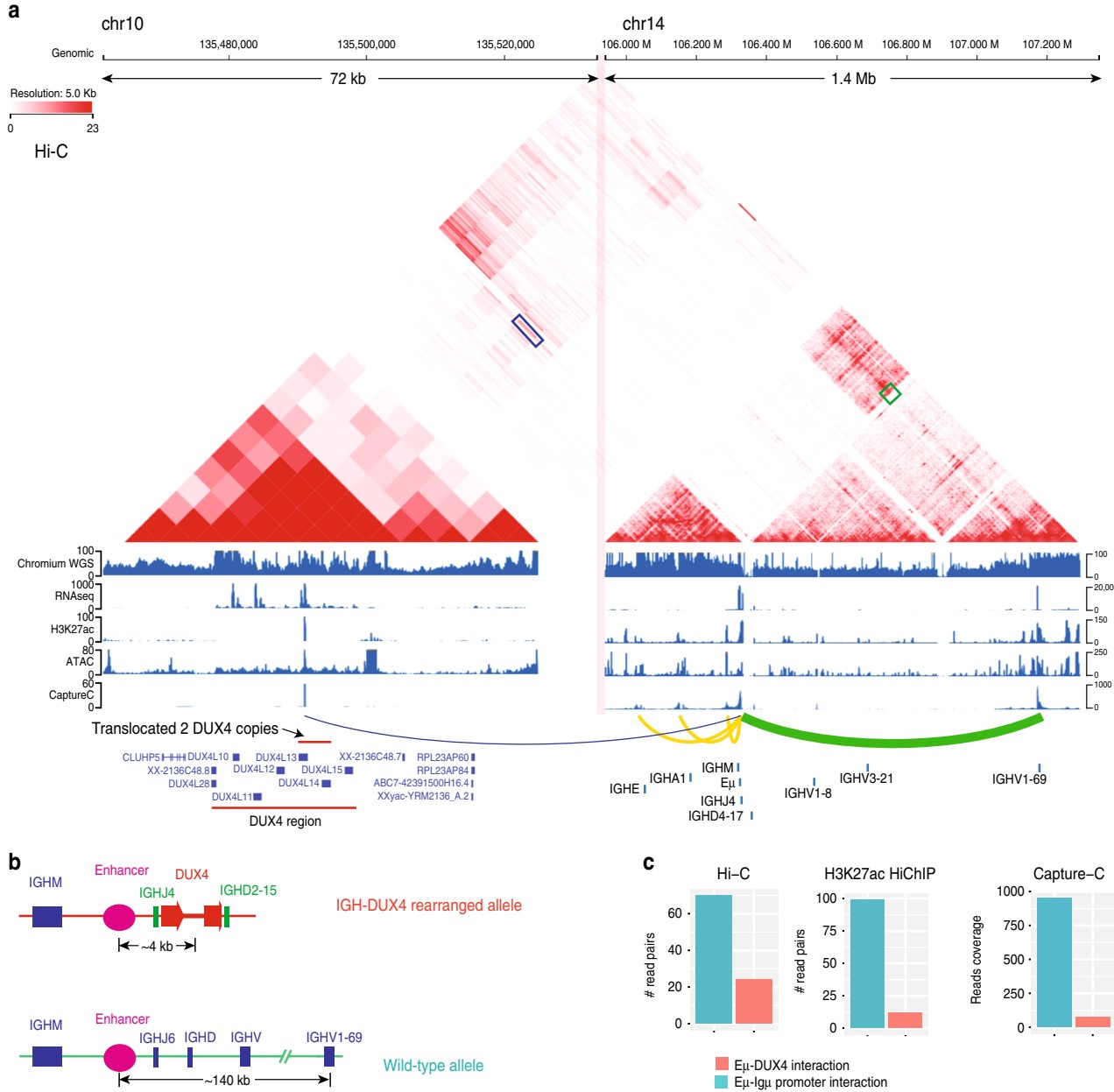

**Fig. 3** Allelic specificity of enhancer–promoter interaction in *IGH-DUX4 haplotype* versus wild-type haplotype in Nalm6. **a** Chromatin interaction plot from Hi-C data at the *DUX4* region (72 kb) on 10q26 and the *IGH* region (1.4 Mb) plotted by the ProteinPaint program[43]. Different *y*-axis scale was used at *DUX4* and *IGH* region for plotting RNA-seq, H3K27ac, ATAC-seq, and Capture-C. The Eμ-*DUX4* interaction and Eμ-Igμ promoter interaction are marked as a blue and green box, respectively. In Capture-C data, the colored arcs represent regions interacting with Eμ, with thickness indicating interaction intensity. Yellow arcs show known Eμ-Igμ interactions. The two translocated *DUX4* copies were marked at *DUX4L13* region, because this region is best matched to our assembled *IGH-DUX4* haplotype. Details are in Methods section. **b** Diagrams of Eμ-*DUX4* interaction and Eμ-Igμ promoter interaction in different alleles. The genomic distances were estimated based on Chromium WGS data and the knowledge that the functional Igμ is from the active *IGH* allele which undergoes VDJ recombination[3]. Details are in Methods section. **c** Comparison of supporting read pairs for Hi-C/H3K27ac HiChIP, the reads coverage for Capture-C between Eμ-*DUX4* interaction and Eμ-Igμ promoter interaction. Multiple mapping issue at *DUX4* regions was considered. The details are described in Methods section

the Pearson's correlation test) between Igu and *DUX4* expression (Fig. 1a). *DUX4* translocation to the silenced allele may thus provide the selective advantage required to achieve the precise level of expression to promote fitness of leukemia cells. In this regard, targeting of *DUX4* by considering the potential for exploiting its oncogenic stress[26,27] may provide a unique therapeutic angle for *IGH-DUX4* B-ALL treatment.

Our analysis of RNA-seq data generated from 54 B-ALL pediatric patient samples with *IGH@* translocation showed that

SV breakpoints on *IGH-DUX4* and *IGH-CRLF2* translocation were highly enriched on *IGH* D-J junctions (red box on Supplementary Fig. 6) while expression of the target oncogenes, i.e., *DUX4* (Fig. 1) and *CRLF2* (Supplementary Fig. 7, data are from published paper[17,28]) is significantly lower than that of Igμ—both patterns match what we observed in Nalm6. The consistency between patient data and the Nalm6 cell line suggest that *IGH@* proto-oncogene translocation on the silenced allele, discovered in Nalm6 through comprehensive analysis on haplotype structure

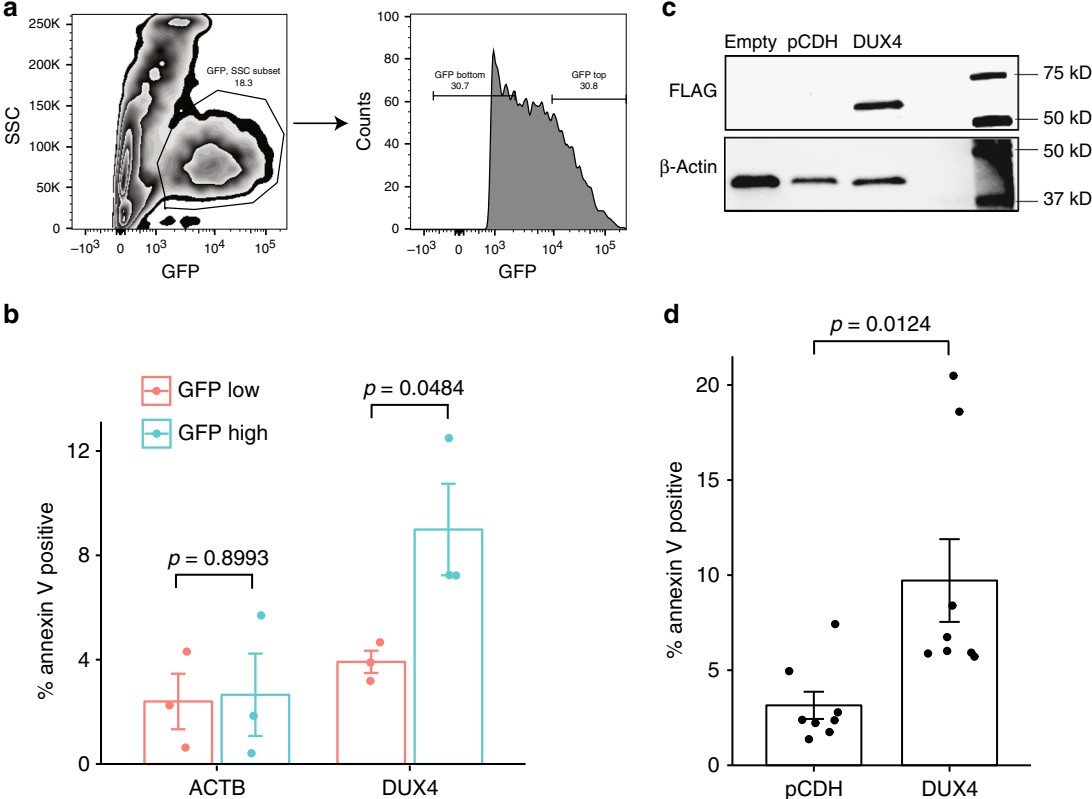

**Fig. 4** *DUX4* overexpression leads to an increased apoptosis. **a** Murine bone marrow cells were transduced with a retroviral vector containing *DUX4* or beta actin (*ACTB*) as a control. Apoptosis was measured by Annexin V staining. Gating strategy is shown. GFP positive cells were gated on, the top 30% and bottom 30% mean fluorescence intensity (MFI) were selected and analyzed for annexin V staining. **b** Percent apoptotic cells as determined by annexin V staining in GFP low and GFP high murine bone marrow populations are shown. Data are combined from three independent experiments. **c** Nalm6 cells were transduced with a lentiviral vector containing GFP alone (pCDH) or FLAG tagged *DUX4* IRES GFP (*DUX4*). Transduced cells were flow sorted for GFP positive populations followed by protein extraction and western blot analysis. Non-transduced cells were included as a control (Empty). Blots were stained using an anti-FLAG antibody (FLAG) and anti-beta actin (β-actin). Sorted cells from three separate experiments were pooled for the western blot. **d** Apoptosis was measured by Annexin V staining in flow sorted GFP positive cells. Data are combined from eight independent experiments. **b**, **d** Two-tailed *p* values calculated using the unpaired *t*-test are indicated. All data are represented as means with standard error of the mean

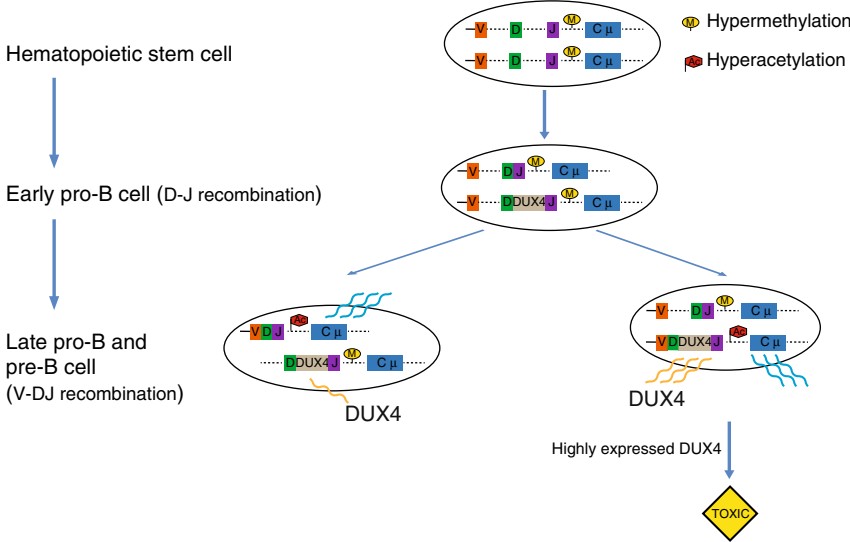

**Fig. 5** A proposed model for the interplay between *IGH-DUX4* translocation and *IGH* allelic exclusion during B-cell development

and epigenetic profiling, could also occur in B-ALL patient samples. Further studies employing haplotype analysis and epigenomic profiling using patient samples or patient-derived xenograft mouse models are needed to provide additional support for this hypothesis.

## Methods

**Evaluation of gene expression level from RNA-seq.** RNA-seq data (GSE85632) for human cleavage cells[13] were downloaded from the GEO database by SRA toolkit v2.8.2-1. The data were aligned to hg19 (GRCh37-lite) using STAR v2.5[29]. Published RNA-seq data of 32 *IGH-DUX4* and 24 *IGH-CRLF2* B-ALL patients were used for the analysis[9,17,28], which were available in EGA (EGAS00001001923, EGAS00001000654) and dbGaP (phs000218). FPKM was calculated for each gene based on the gene models in GENCODE v19 using HTseq-count[30]. Since there are many copies of *DUX4* in the human genome, in order to evaluate *DUX4* gene expression level, all the reads were mapped to DNA segment of a single *DUX4* locus (*DUX4L13* 1267 bp length, chr10:135,490,575–135,491,841 in hg19), then the mapped read counts were used to calculate FPKM. Our local assembly of *IGH-DUX4* haplotype derived from the Chromium WGS showed *DUX4L13* on the reference genome is the best match to the translocated *DUX4*. For Igμ, different B cells use different V/D/J components; however, the constant region of Igμ (*IGHM*) is used in all pre-B cells[1]. The FPKM of *IGHM* was therefore used to evaluate Igμ gene expression.

**10x Genomics Chromium WGS in Nalm6 and haplotype phasing.** High molecular weight genomic DNA of Nalm6 (ATCC, Manassas, VA, USA; the same origin below) was extracted using Qiagen MagAttract HMW DNA Kit. 1.0 ng HMW gDNA was input, and each molecule was captured by a gel bead which contained specific barcode to distinguish different molecules on 10X Genomics Chromium Controller, then amplified by isothermal incubation. The barcoded DNA was purified, then end repaired, dA-tailed and adaptor ligated using chromium genome library kit. Finally, the library was purified and enriched by index PCR amplification then sequenced paired-end 151 cycles on Illumina HiSeq4000.

The Long Ranger 2.0.0 pipeline (https://support.10xgenomics.com/genome-exome/software/pipelines/latest/using/wgs) was used to perform variant calling and phasing. First, we mapped the reads to hg19 with Lariat using default parameters. Second, Freebayes v0.9.21 was used to call haplotype-based variants. Finally, we filtered the variants using bcftools v1.1 (-e '(%QUAL<= 15 || (AF[0]>0.5 && %QUAL<50))' and -e '(AO[0]<2 || AO[0]/(AO[0]+RO) <0.15)').

**Local assembly of *IGH-DUX4* translocation DNA.** The Chromium molecular barcodes were used to select reads near the *IGH-DUX4* rearranged region. First, the 10X Chromium WGS reads mapped to the *IGH* region (chr14:106,000,000–107,349,540 in hg19, from *IGHA* to the chromosome end) were selected. Second, we obtained the barcodes from the selected reads, only keeping barcodes with ≥ 20 reads. Third, all the reads with these barcodes in the whole library were selected, even where the read was unmapped or was mapped to non-*IGH* regions. Fourth, de novo DNA assembly was performed based on the selected reads using Supernova v1.1.4[31] with parameter "output_pseudohap". Fifth, all the assembled contigs were mapped to GRCh38 using blat[32]. We used GRCh38 for this analysis because the annotation on *IGH* and *DUX4* is much improved on this more recent version of the reference genome. An 8063 bp contig was mapped to the *IGHM* region (marked as a blue bar in Supplementary Fig. 1a). Yasuda et al.[8] reported DNA sequences near the *IGH-DUX4* breakpoints in Nalm6 (marked as green bar in Supplementary Fig. 1a) using PacBio sequencing. However, no contig was found to link *IGHM* and the known sequence at the rearranged region. A possible reason is that a 3.5 kb tandem repeat sequence existed in this linking region, making it difficult to fully assemble the entire region. Therefore, the reference genome sequence (chr14:105857149–105863252 in hg38) was used in this linking region (marked as a yellow bar in Supplementary Fig. 1a). We also found a 4637 bp contig span encoding *DUX4* and the intergenic region between two *DUX4* repeats. Based on this contig and the known sequences near breakpoints, the sequence of translocated *DUX4* was manually constructed (marked as red bar in Supplementary Fig. 1a); the length (4670 bp) was consistent with that reported by Yasuda et al.[8] (4663 bp). In summary, we manually assembled a 23,361 bp DNA segment, including *IGHM*, the *IGH* intronic enhancer (Eμ), the translocated *DUX4* sequence, and the sequence near the breakpoints.

We mapped the ~23 kb sequence to hg19 and found the assembled *DUX4* region was best matched to *DUX4L13*. *DUX4L13* showed exactly the same as our assembled sequence. The other *DUX4* copies in hg19 showed at least one mismatch with the assembled sequence.

**Iso-Seq and data analysis.** One microgram of total RNA per reaction was reverse transcribed using the Clontech SMARTer cDNA synthesis kit. Four reverse transcription (RT) reactions were processed in parallel. PCR optimization was used to determine the optimal amplification cycle number for the downstream large-scale PCR reactions. A single primer (primer IIA from the Clontech SMARTer kit 5′-

AAGCAGTGGTATCAACGCAGAGTAC-3′) was used for all PCR reactions post-RT.

Large-scale PCR products were purified with either 1× or 0.4× AMPure PB beads (1× or 0.4× cDNA library hereafter) and the bioanalyzer was used for QC. Equimolar ratios of the 1× and 0.4× cDNA libraries were pooled to generate the non-size-selected SMRTBell library. A total of 3.5 μg of 1× cDNA library was input into size fractionation using the Sage BluePippin system. A 4.5–10 kb size fraction was eluted, re-amplified and cleaned with AMPure PB beads. A SMRTBell library was constructed using 2.3 μg of size-selected cDNA (https://www.pacb.com/wp-content/uploads/Procedure-Checklist-Iso-Seq%E2%84%A2-Template-Preparation-for-Sequel-Systems.pdf).

The non-size-selected and size-selected Iso-Seq SMRTBell libraries were pooled in a 3:1 ratio and co-loaded on a single Sequel cell. A total of two SMRT cells (one 6 h and one 10 h movie) were sequenced on the PacBio Sequel platform using 2.0 chemistry.

After running Iso-Seq pipeline, 144,756 polished circular consensus sequences (CCS) reads (201–14,703 bp) were generated. Then all the reads were mapped to the assembled DNA segment using STAR v2.5[29] with recommended parameters for Iso-Seq (https://github.com/PacificBiosciences/IsoSeq_SA3nUP/wiki/Old-Tutorial:-Optimizing-STAR-aligner-for-Iso-Seq-data). The majority of the mapped reads contained *IGHM* exons (Supplementary Fig. 1b) and *IGHV*1–69 (not shown, because it is excluded in the assembled sequence), and these reads were in-frame coding for functional Igμ. Two *IGH-DUX4* fusion transcripts were found, and the shorter sequence was supported by 3′ RACE experiment by Yasuda et al.[8]. Two antisense *DUX4* transcripts containing *IGHM* exons were also identified (Supplementary Fig. 1b).

**RNA-seq/H3K27ac ChIP-seq/ATAC-seq in Nalm6 and data analysis.** Total stranded RNA-seq in Nalm6 were from our previously published work[9] and available in EGA (EGAS00001000654).

For ATAC sequencing, we followed the protocol established by Buenrostro et al.[33]. Briefly, around 50,000 Nalm6 cells were used for nuclei preparation. Chromatin fragmentation and library construction were carried out using the Nextera DNA sample preparation kit (Illumina, Cat# FC-121-1030). DNA library sequencing was done on an Illumina Hiseq 2500 with 75 bp paired-end reads. The reads were trimming for Nextera adapter by cutadapt (version 1.9, paired-end mode, default parameter with " -m 6 -O 20") and aligned to human genome hg19 (GRCh37-lite) by BWA (version 0.7.12-r1039, default parameter)[34], duplicated reads were then marked with Picard (version 2.6.0-SNAPSHOT)[35] and only non-duplicated proper paired reads have been kept by samtools (parameter "-q 1 -F 1804" version 1.2)[36]. After adjustment of Tn5 shift (reads were offset were offset by + 4 bp for the sense strand and −5 bp for the antisense strand) we separated reads into nucleosome free, mononucleosome, dinucleosome, tri-nucleosome as described in Buenrostro et al.[37] by fragment size and generated bigwig files by using the center 80 bp of fragments and scale to 30 M nucleosome free reads. We observed reasonable nucleosome free peaks and pattern of mono-, di-, tri-nucleosome on IGV (version 2.4.13)[38]. The samples have more than 20 M nucleosome free reads so we conclude the data qualities are good and have enough depth to not miss any chromatin accessible regions.

For the H3K27ac ChIP-seq, a frozen cell pellet containing 10 million cells was sent to Active Motif for ChIP and library preparation. The sample was divided into an aliquot for ChIP using an antibody to H3K27ac (Active Motif) and an input control. Single-end sequencing was performed using an Illumina NextSeq 500 generating 76 cycles for each sequencing read. The reads were mapped to human genome hg19 (GRCh37-lite) by BWA (version 0.7.12-r1039, default parameter)[34], duplicated reads were then marked with Picard (version 2.6.0-SNAPSHOT)[35] and only non-duplicated reads have been kept by samtools (parameter "-q 1 -F 1024" version 1.2)[36]. We followed ENCODE guideline[39] for quality control. Briefly, we used SPP[40] to draw cross-correlation plot and calculated relative strand correlation value (RSC). We then estimate the fragment size from cross-correlation plot and extended each read to the estimated fragment size to generate bigwig files (normalized to 15 million unique mapped reads). Observed clear peak shape along with RSC > 1 and ~40 M (ENCODE criterion 10 M) unique mapped reads, we conclude our IP sample for H3K27ac is good.

**Whole-genome bisulfite sequencing in Nalm6 and data analysis.** In total, 200 ng genomic DNA of Nalm6 was sheared around 400 bp using Covaris LE220 and followed by bisulfite conversion using EZ DNA Methylation-Gold Kit. The fragmented, bisulfite-converted single-strand DNA was tailed and ligated of truncated adapter one to 3′ end, primer extended and truncated of adapter two at the bottom strand only. Finally, the library was purified and enriched by six cycles of index PCR amplification then sequenced paired-end 151 cycles on Illumina NovaSeq 6000.

The reads were mapped to hg19 using bsmap v2.74[41] (-q 20 -u -v 0.08 -w 100 -z 33). The mapping rate was 92.9%, and the bisulfite conversion rate was 99.3% in mitochondria. For allele-specific DNA methylation analysis, in *IGHM* region, only the heterozygous SNPs rs1136534 was in CG island. All 9 CG positions near the SNP (left and right 45 bp region) were considered. Because the SNP rs1136534 located in CG content, only the reads with meaningful orientation overlapped with the SNP were considered. For each selected read, we first identified if the read was a

wild type or *IGH-DUX4* rearranged allele before assessing the number of methylated or unmethylated reads to define coverage and the methylation ratio at the 9 CG positions.

**Hi-C and data analysis.** The Nalm6 cell line was cultured under recommended conditions to about 80% confluence. Five million cells were crosslinked with 1% formaldehyde for 10 min at room temperature, then digested with 125 units of MboI, and labeled by biotinylated nucleotides and proximity-ligated. After reverse crosslinking, ligated DNA was purified and sheared to 300–500 bp, then ligation junctions were pulled down with streptavidin beads and prepared as general Illumina library. The Hi-C sample was sequenced paired-end 76 cycles on Illumina Hiseq 4000.

The reads were mapped to hg19 and processed using Juicer v1.5[42] with default parameters. Of ~2.5 billion Hi-C reads, 60.9% were Hi-C contact reads. Because many copies of *DUX4* exist in the reference genome, in order to show the reasonable signals in the *DUX4* region, no minimum mapping quality was required. The data matrices were further extracted from .hic files with the dump function from juicer tools, and the data were visualized with the Hi-C viewer under development though the Pediatric Cancer (PeCan) data portal[43].

The *IGH* intronic enhancer (Eμ) lies between the tandem repeat region and *IGHJ* region[44]. In order to extract the contact reads between Eμ and Igμ promoter from Hi-C data, we defined Eμ and Igμ promoter as the regions with highly enriched H3K27ac/ATAC-seq reads, marked as red boxes in Supplementary Fig. 8a and 8b. Only the read pairs that one read falls in Eμ and the other in the Igμ promoter were used to evaluate the Eμ-Igμ promoter interaction. Because of multiple mapping issue for *DUX4* region, to quantify Eμ-*DUX4* interaction reads, all the read pairs that one read falls in Eμ and the other in any *DUX4* array region annotated by Gencode V28lift37 were defined as the read pairs supported Eμ-*DUX4* interaction.

**H3K27ac HiChIP and data analysis.** The Nalm6 cell line was cultured under recommended conditions to about 80% confluence. DNA from 10 million cells were crosslinked with 1% formaldehyde for 10 min at room temperature, then digested with 375 units of MboI, labeled by biotinylated nucleotides and proximity-ligated. After being sheared by Covaris, chromatin immunoprecipitated (ChIP) with H2K27ac antibody, the ChIP DNA was de-crosslinked and purified by Zymo Genomic GNA Clean & Concentrator kit. Finally, after streptavidin beads pull-down and Illumina library construction by Kapa Hyper Prep Kit, hiChIP library was sequenced paired-end 76 cycles on Illumina NextSeq 500.

HiC-Pro[45] was used for H3K27ac HiChIP data analysis. Of ~371.6 million reads, 75.6% were contact read. The same defined enhancer/promoter region and the same strategy for Hi-C data analysis were used to quantify Eμ-*DUX4* interaction reads and Eμ-Igμ promoter interaction reads in H3K27ac HiChIP data.

**Capture-C and data analysis.** Capture-C was designed to identify the regions interacting with Eμ. There were three DpnII cutting sites (shown as green bars in Supplementary Fig. 9) in the enhancer region. Three oligos (Supplementary Table 1) were designed near the cutting sites to capture Eμ. Nalm6 was cultured to about 80% confluence and resuspended at a concentration of 10e7 cells in 10 ml of medium. Cells were crosslinked with 2% formaldehyde and quenched by 0.125 M glycine. After lysis, cells were kept on ice for 20 min, digested with 500 units of DpnII, then underwent the proximity ligation. After reverse crosslinking, ligated DNA was purified, then sheared to 200 bp and prepared as general indexed Illumina libraries. Two micrograms of different indexed samples were captured twice by specific biotinylated oligos, then PCR amplified streptavidin beads pulled down libraries. The NG capture-C libraries were sequenced on Miseq paired-end 151 cycles.

The data were analyzed using the pipeline capture-C[46]. In order to avoid very high read coverage in the oligo region during signal visualization, the reads at the extended 500 bp region of the fragment (shown as a red bar in Supplementary Fig. 9) were filtered. In addition to the interaction of Eμ-Igμ promoter and Eμ-*DUX4*, interactions were also detected between Eμ and *IGHA/IGHE* (close to *IGH* 3′ enhancer) (Fig. 3a). This result was consistent with current knowledge that Eμ is also in contact with the region near *IGH* 3′ enhancer[47,48].

The highest peak at *DUX4* region and Igμ promoter region were used to quantify Eμ-*DUX4* and the Eμ-Igμ interaction. Although many *DUX4* copies exist in human genome, only one peak of 56 reads at *DUX4L13* (the exact same sequence as the *DUX4* sequence on our assembled *IGH-DUX4* haplotype in Nalm6) was found (shown in Fig. 3a). The enrichment of Capture-C mapping to *DUX4L13* can be explained by its ≥1-bp difference with all other annotated *DUX4* genes on hg19. To account for multiple mapping issue, we identified a total of 101 reads mapped to any *DUX4* clusters based on the annotation of Gencode V28lift37 and mapped these 101 reads to the *DUX4L13* nearby region (chr10:135,488,523–135,493,885 in hg19) using BWA[34], resulting in an increase of *DUX4L13* peak coverage to 75. Therefore, the fold change of interaction intensity between the Eμ-Igμ promoter and Eμ-*DUX4* was estimated to be 13 fold (953 vs. 75).

**Genomic distance between Eμ and Igμ promoter/*DUX4*.** From our assembled sequence of *IGH-DUX4*, the linear distance between Eμ and *DUX4* is ~4 kb (Fig. 3b). Because the allele with Igμ expression undergone VDJ recombination, most *IGH* V regions showed one allele deletion from Chromium WGS. However, a ~126 kb V region was maintained in both alleles (Fig. 3a). Therefore, we estimated the linear distance between Eμ and Igμ promoter is ~140 kb (Fig. 3b).

**A reciprocal translocation between chr10 subtelomere and *IGH*.** A large-scale Hi-C plot between chr10 subtelomere (1.4 Mb) and chr14 *IGH* (1.4 Mb) was shown in Supplementary Fig. 5a. The translocation event "2 *DUX4* copies were inserted into *IGH* D and J region" was shown as blue box. Most *IGHV* regions were deleted on the wild-type *IGH* allele based on well-known allelic exclusion of *IGH*[3]. From the coverage of Chromium WGS, the big *IGHV* region "C" (marked as red bar) has one allele deletion, suggesting sequence at region "C" is on the non-wild-type *IGH* allele. The region "C" does not interact with *IGHM*/enhancer region "A" (marked as green bar) but interact with chr10 subtelomere (marked as cyan bar) shown as yellow box, suggesting a reciprocal translocation between chr10 subtelomere and *IGH* region. A model of the two translocation events between chr10 subtelomere and the non-wild-type (alternative) *IGH* allele was shown in Supplementary Fig. 5c.

**Apoptosis assay.** For the murine experiments, full-length *DUX4* and *ACTB* cDNA were cloned into the MSCV-IRES-eGFP retroviral vector. Ecotropic envelope-pseudotyped virus was generated using the Phoenix™-Eco cell line (Thermo Fisher Scientific, Waltham, MA, USA) as per manufacturers instructions. Murine bone marrow from 4- to 6-week-old C57BL/6 mice was harvested, lineage depleted as per manufacturers instructions (Miltenyi Biotec, Bergisch Gladbach, Germany), and cultured in the presence of murine IL-3 (10 ng/ml), IL-6 (30 ng/ml), and SCF (50 ng/ml) (Pepro Tech US, Rocky Hill, NJ, US) for 24 h prior to transduction on RetroNectin (Takara Bio Inc.)[49]. Thirty-six to forty-eight hours following transduction with a single hit of virus, cells were harvested and analyzed by flow cytometry. Briefly, cells were washed and stained with annexin V conjugated to APC (BD Biosciences, San Jose, CA, USA) for 15 min at 4 °C, washed and resuspended in 4′,6-diamidino-2-phenylindole containing solution (1 μg/ml DAPI) for analysis using FACS LSR II D (BD Biosciences, San Jose, CA, USA). All animal studies were maintained and treated in accordance with guidelines approved by the Institutional Animal Care and Use Committee (IACUC) at St. Jude Children's research hospital. For the Nalm6 cell line, a FLAG-tagged full-length *DUX4* gene was cloned into the lentiviral construct pCDH (System Biosciences, Palo Alto, CA, USA). VSV-G pseudotyped virus was generated by transient transfection as per manufacturer's instructions (System Biosciences, Palo Alto, CA, USA). Thirty-six hours following transduction with a single hit of virus, cells were harvested and analyzed by flow cytometry and described above. GFP-positive populations were flow sorted for western blot analysis to confirm expression of *DUX4* from the lentiviral construct. The details of gating strategy for mice bone marrow and Nalm6 can be found in Supplementary Fig. 10.

In *IGH-DUX4* B-ALL patients, the C-terminal truncation of *DUX4* caused by translocation varies significant, from 4aa to ~200aa (16aa for Nalm6) based on previous studies by us and others[8–10] (https://pecan.stjude.cloud/proteinpaint/DUX4). In a recent study, Dong et al. found that the wild-type *DUX4* and *IGH-DUX4* share similar DNA binding mechanism[10]. The crystal structure of Apo-DUX4HD2 and DNA-bound complex suggests that the N-terminal double homeobox domains are essential for *IGH-DUX4* driven transactivation and B-ALL leukemogenesis. Combined, these data suggest the enhancer hijacking leading to expression of the *DUX4* peptide is the real oncogenic effect. Therefore, the wild-type *DUX4* was utilized in our apoptosis experiments.

**Western blot analysis.** Total protein was extracted from GFP positive flow sorted Nalm6 cells with RIPA lysis buffer and blotted with anti-FLAG (Sigma F3165; 10 μg/ml) and anti-Beta Actin (Abcam ab8227; 1:1000 dilution). Anti-mouse IgG superclonal-HRP (Invitrogen A28177; 1:5000 dilution) and anti-rabbit-HRP (Abcam ab6721-1; 1:3000 dilution) were used as secondary antibodies for FLAG and Beta Actin antibodies, respectively. Uncropped and unprocessed scans can be found in Supplementary Fig. 11.

**Reporting summary.** Further information on research design is available in the Nature Research Reporting Summary linked to this article.

## Data availability

10X Chromium WGS, PacBio Iso-Seq data, and bisulfite sequencing data at *IGHM* region are available in NCBI BioProject PRJNA473990. H3K27ac ChIP-seq, Hi-C, HiChIP, Capture-C, and ATAC-seq data are available in the GEO database (GSE115494).

## Code availability

The codes used in the analysis are available at https://github.com/liqingti/IGH-DUX4.

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

## Acknowledgements

We thank Shuoguo Wang and Ti-Cheng Chang for help with 10X Chromium WGS analysis, and Drs. Hongbo Chi and Yongqiang Feng for critical review of the manuscript. We thank Lily Maxham and Janet Partridge for confirming DUX4 toxicity experiment using inducible expression system in Nalm6. This research was supported by the National Cancer Institute of the National Institutes of Health under Award Number R01CA216391. The content is solely the responsibility of the authors and does not necessarily represent the official views of the National Institutes of Health. This research was also supported by Cancer Center Support Grant P30CA021765 from the National Institutes of Health and in part by the American Lebanese Syrian Associated Charities (ALSAC).

## Author contributions

J.Z. conceived the project. J.Z. and L.T. designed the study. L.T., B.X., X.M., Y.L., P.S., and X.Z. analyzed the data. S.Newman provided data analysis advice. Y.S., B.J., and J.E designed and performed 10X Chromium WGS/ATAC-seq/Hi-C/HiChIP/Capture-C experiment. E.T, T.H., and M.A. performed Iso-Seq data. T.A.G. designed the apoptosis experiments which were performed by S.Nance and J.D. L.D. helped on western blot experiment. C.L. advised and gave support to the experiment. All authors contributed to the writing, editing, and completion of the manuscript.

## Additional information

**Competing interests:** E.T., T.H., and M.A. are employed by Pacific Biosciences. The other authors declare no completing interests.

