## [Transparent Peer Review File · Nature Communications]

Reviewers' comments:

Reviewer #1 (Remarks to the Author):

The authors present a nice integrative analysis revealing that DUX4 is translocated to the silenced IGH allele. While the authors use a variety of methods to come to this finding, the underlying biology of why DUX4 is overexpressed by just the silenced allele (compared to fusion negative patients) and not overexpressed by the other allele could be pursued further. The authors raise the hypothesis "that the IGH-DUX4 translocation is exclusively expressed on the silenced haplotype because higher expression levels of DUX4 from an active E μ cannot be tolerated without triggering apoptosis." Further experimental work demonstrating this would be needed to warrant publication in Nature Communications.

Reviewer #2 (Remarks to the Author):

The authors integrated several high-throughput technologies, including long-read iso-seq, RNA-seq, Hi-C, ATAC-seq etc to elucidate the molecular basis of IGH-DUX4. They addressed an interesting finding that DUX4 fused with the silenced allele of IGH resulted in decreased DUX4 expression. However, the evidence is weak and more evidence is required to strengthen the argument.

Major points:

1. The author found that the silenced IGH allele was fused to DUX4. How about the expression of the fused DUX4 compared with wild-type DUX4? What will happen if the enhancer/promoter activity of the silenced IGH was higher than the enhancer/promoter activity of DUX4? In addition to the comparison of two alleles of IGH, the author should also compare the expression/SNP/Hi-C/histone modification on the two alleles of DUX4.
2. The author claimed that the major benefit of rearrangement of IGH with DUX4 was the resultant low expression of DUX4, which suggested that the IGH-DUX4 or DUX4 overexpression might be an unfavorable factor during leukemogenesis. However, lines of evidence have demonstrated that IGH-DUX4 is essential in B-cell leukemia and overexpression of IGH-DUX4 drives B-cell leukemia in vivo (Nat Genet. 2016 May; 48(5):569-74; Leukemia. 2018 Jun; 32(6):1466-1476.). To address the conclusion that translocation of the silenced IGH resulted in reduced expression of DUX4, the author should provide evidence that 1) overexpression of DUX4-IGH was toxic in NAML6, or 2) DUX4 knockdown could benefit B-cell leukemia.
3. The author found that IGH fused to DUX4 was silenced and the silenced IGH lacked H3K27ac binding, ATAC-seq signal, and the enhancer-promoter interaction. What is the mechanism accounting for the imbalanced histone modification or 3D structure on the IGH loci?
4. The author should provide evidence on the silenced IGH of IGH-DUX4 in the primary sample. For example, the author might perform the WGS/Hi-C/H3K27ac ChIP-seq/ATAC-seq in the primary sample with the IGH-DUX4 fusion and see if DUX4 was indeed fused to the silenced allele of IGH.

Minor

1. Why are the SNPs used in Fig 1 and Fig 2 different?
2. Fig 1a. The authors had better to compare the expression of IGH-fusion and wild-type DUXs, instead of wild-type IGH.
3. DUX4 is mapped using hg38 while the expression DUX4 is measured using hg19. Please unify.

Reviewer #3 (Remarks to the Author):

The article entitled « Long-read sequencing unveils oncogenic translocation to silenced IGH allele in pediatric leukemia », by Tian et al. investigates the IGH-DUX4 translocation in a cell line by several novel techniques (long read genome profiling, transcriptome sequencing, epigenetic and 3D profiling) and concludes that this translocation occurs on the silenced IGH allele. Although the molecular data appear convincing, some of the conclusions are not supported by the data.

Why is it important to know that translocation can occur on the silenced copy? Are there medical or biological consequences and application to this observation? Only a weaker transactivation?

The authors use RNA-seq data to evaluate the expression level. RNA level reflect transcription but also RNA stability. To assure the comparison, could the author show that DUX4 RNA stability is comparable to the stability of the functional IGH RNA.

If one can agree that the translocated copy of the IGH locus is less expressed than the IGH copy carrying the functional rearrangement, this does not imply that the translocation occurred on a previously silenced copy. The translocation may occur on a copy and, at the end of the transformation process, the translocated locus ends up less transcribed than the one carrying the functional IGH rearrangement. So the authors do not provide any proof that the translocation occurs on a silenced copy. Furthermore, no reason is provided to extend the conclusion to other B-ALL translocations based only on the location of the breakpoints in the D-J regions. Line 110-114: I do not understand the rationale: breakpoints at D and J regions are enriched for DUX4 and CRLF2, and so what ? please clarify

What is the range and diversity of IGH copy expression in B-ALLs? Is the expression ratio between functional/nonfunctional copy different between IGH-translocated and non translocated samples? Is there any hint on the differentiation steps those samples are arrested at, with respect to normal B-cell development?

SupFig4 : The used vectors need to be detailed, and the levels of expression of the DUX4 proteins could be checked. What are the transduced cells? Bone marrow cells are a mix of cells. Do they represent the cells in which the translocation occurred? Or at minimal, do they represent the tumor cells?

Are the Nalm6 cells diploid? Do they have only one copy of each IGH locus?

Minor remarks

Line 34: between

Line 37: please precise where does the DUX4 gene lies ? : chromosome 4 (4q35 ?), chromosome 10 ? where ?

Line 46 : please explain what means FPKM

Line 112 : how many patients

Figure1. « only exonic SNPs » : please precise which ones

Figure 2a and sup4 : please explain why chromosome 10, and not chromosome 4. Please explain what means ACTB

Reviewer #4 (Remarks to the Author):

In this study an extensive range of state-of-the-art technologies is used to analyze the haplotypes

involved in the IgH-DUX4 translocation present in the Nalm6 B-ALL cell line. The authors clearly demonstrate that it is the silenced IgH allele that is fused to the DUX4 locus, which suggests that DUX4 upregulation in the Nalm6 cell line is the result of hijacking the E μ enhancer on the silenced IgH allele. In an attempt to extend this observation made in a single cell line, cells from B-ALL patients carrying an IgH@ translocation are taken and analyzed for the expression levels of the translocation partner genes. These are lower than the levels observed for Ig μ in these patients, which is interpreted as support for a common mechanism in B-ALL in which the silenced IgH allele is selected for translocations.

Except for some minor comments below, I find this a technically well performed study. However, for reasons clarified below, I have doubts about the biological significance of these findings. I also have concerns about the authors' interpretation of differences in expression levels.

1. I am not an expert in B cell development, but is it possible that in order to become a B-cell with acute lymphoblastic leukemia (B-ALL) properties, successful rearrangement and production of IgH (Ig μ in case of the Nalm6 B-ALL cell line used here) is a prerequisite, otherwise you are not a B-cell? If so, to me this would imply that in B-ALL it will always be only the non-productively rearranged (silenced) IgH allele that is available for rearrangements to an oncogene and that the findings presented here would be entirely unsurprising? Authors, please discuss this issue.

2. The authors state in their introduction "Intriguingly, DUX4 expression was much lower than Ig μ in all 32 patients examined (the median FPKM of DUX4 is only 21% of that of Ig μ , Fig. 1a), raising the alternative possibility that IGH-DUX4 translocation might occur on the silenced IGH allele". Later in the text (line 114) they again compare expression levels of Ig μ to that of translocation partner genes, to draw a similar conclusion: In IGH-DUX4 patients, we found lower expression of DUX4 than Ig μ (Fig. 1a). The same pattern was also found in IGH-CRLF2 patients— the median FPKM of CRLF2 is only 26% of that of Ig μ from published RNA-seq data of 24 IGH-CRLF2 patients^{12, 17} (Supplementary Fig. 6). This suggests that oncogenic translocation to the silenced IGH allele could be a common mechanism in B-ALL.

In my opinion, this data cannot be presented as indicative for it being the enhancer on the silenced IgH allele that is responsible for transcription of the translocated gene. After all, expression levels not only depend on enhancer strength but also on the enhancer-promoter combination (we do not know DUX4 levels under the control of the active IgH enhancer: even at its full power, it may still not be able to activate the DUX4 promoter to Ig μ levels!), enhancer-promoter distance, gene length, mRNA stability, etc, etc.

Line 83: To quantify the contact intensity between E μ -DUX4 and the E μ -Ig μ promoter, we designed Capture-C with capture probes in E μ , and found that the contact intensity of E μ -DUX4 was ~14-fold lower than that of the E μ -Ig μ promoter (Fig. 2a). Therefore, the lower expression of DUX4 compared to Ig μ (~15-fold, blue dot in Fig. 1a) likely results from the weaker enhancer-promoter interaction of E μ -DUX4 compared to E μ -Ig μ promoter.

This statement and the analyses raises several questions:

3. It is unclear from Figure 2A where the DUX4 gene starts and ends, i.e. where to look for enhancer-promoter interactions: authors, please properly annotate plots for readers to orient themselves.

4. As the authors acknowledge themselves: many copies of DUX4 exist in the reference genome, making it very difficult to assign mapped reads to a given DUX4 copy. To me this seems to compromise proper quantification of contact frequencies. Authors, please clarify this issue and justify your quantification strategy in the main text.

5. Contacts with DUX4 are less frequent than with the I μ promoter. What is being compared here (what do the boxes represent): contacts across the entire DUX gene versus contacts with the I μ promoter? Please specify and justify the regions that are compared here.
6. The authors use Capture-C data to quantify contact frequencies but they could also do this with the Hi-C data (by windowing the data). Please provide this analysis as well.
7. Contact frequencies with DUX4 are lower: is this because of differences in the linear distances between the E μ - I μ promoter and the E μ -DUX4 promoter? Please specify these distances and discuss if there are differences.
8. The Hi-C data in Figure 2A appear to give lower resolution contact maps at the DUX4 region than at the IgH locus: how is this possible?
9. Please adjust the Capture-C scaling at the DUX4 region (is now 0-500, make it e.g. 0-20). One should see where the rearranged part of the chromosome starts and be able to judge the signal at the DUX4 promoter in the context of that of its immediate surrounding.
10. The author state: 'We hypothesize that the IGH-DUX4 translocation is exclusively expressed on the silenced haplotype because higher expression levels of DUX4 from an active E μ cannot be tolerated without triggering apoptosis'. Please see my first comment: if it is true that in B-ALL only the silenced allele is available for translocations, all such speculations seem irrelevant.
11. For reasons explained in comment 2, I find the observation that expression levels of translocation partner genes are lower than that of I μ in B-ALL patients carrying an IGH@ too circumstantial to be presented as support for translocations with the silenced IgH allele. In my opinion, authors should limit their conclusions throughout the text (including title, abstract) to their carefully analyzed Nalm6 cell line. Alternatively, they should haplotype the translocated chromosomes across a larger panel of B-ALL patients/cell lines.

Reviewer #1 (Remarks to the Author):

The authors present a nice integrative analysis revealing that DUX4 is translocated to the silenced IGH allele. While the authors use a variety of methods to come to this finding, the underlying biology of why DUX4 is overexpressed by just the silenced allele (compared to fusion negative patients) and not overexpressed by the other allele could be pursued further. The authors raise the hypothesis "that the IGH-DUX4 translocation is exclusively expressed on the silenced haplotype because higher expression levels of DUX4 from an active E μ cannot be tolerated without triggering apoptosis." Further experimental work demonstrating this would be needed to warrant publication in Nature Communications.

[Author Response] We appreciate the reviewer's comment on the importance of our finding as well as the need for further experiment to prove oncogenic toxicity of DUX4 in leukemia cells. We acknowledge that in the original submission our experimental data which demonstrated DUX4 toxicity in mouse bone marrow presented was inadequate as DUX4 toxicity in cancer cells was solely based on prior studies on rhabdomyosarcoma cell lines but not leukemia cells.

To evaluate the effects of DUX4 overexpression in fully transformed leukemia cells, we carried out a new experiment to transduce DUX4 into Nalm6 cell line which already carry the IGH-DUX4 translocation. Similar to prior experiment on bone marrow cells, we found that fully transformed leukemia cells did not tolerate further overexpression of DUX4 with a significant increase in apoptosis. The results from both experiments are now presented in the newly included Fig. 4. We therefore conclude that while the level of DUX4 expression caused by DUX4 translocation to the silenced IGH allele is tolerable, increased DUX4 expression expected from DUX4 translocation to the active IGH allele can be toxic to leukemia cells and induce apoptosis. This additional experiment provides a biologic rationale for DUX4 expression from the repressive haplotype.

In the revision, we incorporated the following text in the result: *"To further evaluate the effects of DUX4 overexpression in fully transformed leukemia cells, we introduced DUX4 into Nalm6 cells which already harbor the IGH-DUX4 translocation, confirming GFP-tagged DUX4 protein expression by Western blot (Fig. 4c and 4d). The results shown that fully transformed leukemia cells would not tolerate further overexpression of DUX4 with a significant increase in apoptosis when comparing DUX4 transduced cells with the empty vector control (p=0.0124). We therefore conclude that while low levels of the DUX4 protein are tolerable, high levels induce apoptosis providing a biologic rationale for expression from the repressive haplotype."*

Reviewer #2 (Remarks to the Author):

The authors integrated several high-throughput technologies, including long-read iso-seq, RNA-seq, Hi-C, ATAT-seq etc to elucidate the molecular basis of IGH-DUX4. They addressed an interesting finding that DUX4 fused with the silenced allele of IGH resulted in decreased DUX4 expression. However, the evidence is weak and more evidence is required to strengthen the argument.

[Author Response] We respectfully disagree that our conclusion was reached based on weak evidence. As noted by comments from other reviewers, our analysis on allele-specific expression of long-read RNA-seq as well as allele-specificity of H3K27ac ChIP-seq profiling and chromatin-interaction profiling by Hi-C, Hi-ChIP and Capture C in the Nalm6 leukemia cell line are comprehensive with consistent data supporting translocation of DUX4 into the silenced IGH allele.

Based on the reviewer's comment, we believe further clarification regarding the following two points raised by the reviewer may clarify that we are examining DUX4 expression level from the perspective of its translocation to the silenced versus active IGH allele: 1) DUX4 is transcribed ONLY in embryo during normal development; and 2) DUX4 is NOT expressed in leukemia cells except for those with IGH-DUX4 translocation. These changes have been incorporated into the revised manuscript as described below in our point-by-point response to the reviewer's comments which should clarify that DUX4 is not expressed from its "native" genomic loci in somatic tissue.

Major points:

1. The author found that the silenced IGH allele was fused to DUX4. How about the expression of the fused DUX4 compared with wild-type DUX4? What will happen if the enhancer/promoter activity of the silenced IGH was higher than the enhancer/promoter activity of DUX4? In addition to the comparison of two alleles of IGH, the author should also compare the expression/SNP/Hi-C/histone modification on the two alleles of DUX4.

[Author Response] We would like to clarify that the wild-type DUX4 is not expressed in normal pre-B or mature B cell. As noted in published studies [Hendrickson PG, et al. Nat Genet 49, 925-934 (2017); Geng LN, et al. Dev Cell 22, 38-51 (2012); van der Maarel SM, et al. Trends Mol Med 17, 252-258 (2011)], DUX4 is located within the GC-rich D4Z4 repeats at subtelomeric region that is characterized by high levels of repression, and it is expressed only during events associated with major chromatin relaxation, for instance in early embryos and after loss of repression of the D4Z4 macrosatellite repeat in myoblasts of people with facioscapulohumeral

muscular dystrophy. In B-ALL, DUX4 is expressed only in a specific B-ALL subtype that harbors the IGH-DUX4 translocation as its expression was driven by hijacking IGH enhancer via IGH-DUX4 translocation as described in previous studies by us and the other group [Yasuda T, et al. Nat Genet 48, 569-574 (2016); Zhang J, et al. Nat Genet 48, 1481-1489 (2016); Dong X, et al. Leukemia 32, 1466-1476 (2018)]. Other B-ALL subtypes that lack the IGH-DUX4 translocation do not express DUX4. Therefore, we can not compare the enhancer/promoter of the wild-type versus that of translocated DUX4 gene. To clarify that the wild-type (or more precisely, the “native”) DUX4 is not expressed, we included the following background information of DUX4 in Introduction as follows:

“DUX4 is a transcription factor located within the GC-rich D4Z4 repeat array at the subtelomeric regions of 4q35 and 10q26 which are characterized by high levels of repression^{7, 8}. It is expressed only during events associated with major chromatin relaxation, i.e. in early embryos at cleavage stage (2/4/8-cell embryos)⁹ or after loss of repression of the D4Z4 macrosatellite repeat in myoblasts of individuals with facioscapulohumeral muscular dystrophy (FSHD)^{10, 11}. The IGH-DUX4 subtype accounts for ~7% of the pediatric B-ALLs in which an inter-chromosomal translocation repositions DUX4 within the vicinity of the IGH enhancer (E μ), the native enhancer of IGH gene (Ig μ) in precursor B (pre-B) cell^{4, 5, 12}. IGH-DUX4 translocation is a clonal event acquired early in leukemogenesis of this B-ALL subtype resulting aberrant activation of DUX4 which is absent in other B-ALL subtypes^{4, 5}.”

2. The author claimed that the major benefit of rearrangement of IGH with DUX4 was the resultant low expression of DUX4, which suggested that the IGH-DUX4 or DUX4 overexpression might be an unfavorable factor during leukemogenesis. However, lines of evidence have demonstrated that IGH-DUX4 is essential in B-cell leukemia and overexpression of IGH-DUX4 drives B-cell leukemia in vivo (Nat Genet. 2016 May;48(5):569-74; Leukemia. 2018 Jun;32(6):1466-1476.). To address the conclusion that translocation of the silence IGH resulted in reduced expression of DUX4, the author should provide evidence that 1) overexpression of DUX4-IGH was toxic in NAML6, or 2) DUX4 knockdown could benefit B-cell leukemia.

[Author Response] We thank the reviewer for pointing out the well-established DUX4 oncogenicity discovered by multiple groups including us [Zhang J, et al. Nat Genet 48, 1481-1489 (2016); Yasuda T, et al. Nat Genet 48, 569-574 (2016)]. In the revised manuscript, we also included “Leukemia. 2018 Jun;32(6):1466-1476” cited by the reviewer in Introduction as follows: *“In this study, we investigated allele specificity of IGH-DUX4 translocation, an oncogenic event that defines a B-ALL subtype with a distinct expression profile^{4, 5, 6}.”*

As described in our response to comment #1, DUX4 is NOT expressed in leukemia except for those of the IGH-DUX4 subtype that harbors IGH-DUX4 translocation as reported in our

previous study [Zhang J, et al. Nat Genet 48, 1481-1489 (2016), Fig. 2a]. The current study investigated the allelic-specificity of DUX4 translocation, i.e. whether DUX4 translocation occurred on the silenced versus the active IGH allele and we investigated potential selective advantage for DUX4 translocation to the silenced IGH allele. We hypothesize that oncogenic toxicity, i.e. TOO MUCH of the DUX4 oncogene, may not be good for cancer cells based on the known pro-apoptotic property of DUX4 reported by Kowaljow et al [Neuromuscul Disord 17, 611-623 (2007)] and by the toxicity experiment in mouse bone marrow presented in our original manuscript.

In the revision, we followed the reviewer's suggestion and carried out a toxicity experiment in Nalm6 to provide additional evidence on oncogenic toxicity of DUX4, i.e. too much of oncogene can be "bad" for a tumor cell. We transduced this full-length DUX4 cDNA into the Nalm6 line instead of IGH-DUX4 fusion protein. This is because in patient samples of B-ALL of IGH-DUX4 subtype, the C-terminal truncation of DUX4 caused by translocation varies significantly, from 4aa to ~200aa, based on previous studies by us and others [Zhang J, et al. Nat Genet 48, 1481-1489 (2016); Yasuda T, et al. Nat Genet 48, 569-574 (2016); Dong X, et al. Leukemia 32, 1466-1476 (2018); <https://pecan.stjude.cloud/proteinpaint/DUX4>]. Therefore, enhancer hijacking leading to DUX4 expression is the real oncogenic effect. Most recently, Dong et al. found that the wild-type DUX4 and IGH-DUX4 share similar DNA binding mechanism and their crystal structure of Apo-DUX4HD2 and DNA-bound complex suggested the N-terminal double homeobox domains are essential for IGH-DUX4 driven transactivation and B-ALL leukemogenesis [Dong X, et al. Leukemia 32, 1466-1476 (2018)]. Therefore, using the wild-type DUX4 for the over-expression experiment is appropriate.

The result, presented in the new Figure 4 (4c and 4d), was described as follows in the revision: *"To further evaluate the effects of DUX4 overexpression in fully transformed leukemia cells, we introduced DUX4 into Nalm6 cells which already harbor the IGH-DUX4 translocation, confirming GFP-tagged DUX4 protein expression by Western blot (Fig. 4c and 4d). The results shown that fully transformed leukemia cells would not tolerate further overexpression of DUX4 with a significant increase in apoptosis when comparing DUX4 transduced cells with the empty vector control (p=0.0124). We therefore conclude that while low levels of the DUX4 protein are tolerable, high levels induce apoptosis providing a biologic rationale for expression from the repressive haplotype."*

DUX4 knockdown in Nalm6 has already been carried out by Yasuda et al [Nat Genet 48, 569-574 (2016)], which shown that knockdown of the IGH-DUX4 fusion suppressed the proliferation of Nalm6 cells. Therefore, there is no need to repeat this experiment in our study. The result from the knockdown experiment (by Yasuda et al) coupled with the oncogenic toxicity observed by the over-expression in Nalm6 in the revised manuscript make a good case for suggesting that DUX4 expression in Nalm6 follow the "Goldilocks principle"—i.e. "just-right" levels are required as too-much will lead to apoptosis and too-little will lead to suppression of

proliferation as described in an review article on oncogene overdose by Dipak Amin et al [Biomark Cancer. 2015 Dec 15].

In the revised manuscript, we incorporated the previous DUX4-knockdown experiment along with the DUX4-overexpression toxicity experiment in the Discussion as follows:

“Previously, Yasuda et al. has shown that knockdown of the IGH-DUX4 fusion would suppress proliferation of Nalm6 cells⁴, confirming the oncogenic potential of DUX4 expression in B-ALL. This, coupled with the oncogenic toxicity caused by overexpression of DUX4 presented in this study (Fig. 4), suggests that DUX4 expression in Nam16 may follow the “Goldilocks principle”²⁶ — i.e. “just-right” levels are required as too-much will lead to apoptosis while too-little will lead to suppression of proliferation. DUX4 translocation to the silenced allele may thus provide the selective advantage required to achieve the precise level of expression to promote fitness of leukemia cells.”

3. The author found that IGH fused to DUX4 was silenced and the silenced IGH lacked H3K27ac binding, ATAC-seq signal, and the enhancer-promoter interaction. What is the mechanism accounting for the imbalanced histone modification or 3D structure on the IGH loci?

[Author Response] We thank the reviewer for this comment. We know from previous studies that the mechanism responsible for the imbalanced histone modification/state between the two IGH loci involves a stepwise epigenetic process which has been reported in a number of studies [Bergman Y, Cedar H. *Nat Rev Immunol* **4**, 753-761 (2004); Vettermann C, Schlissel MS. *Immunol Rev* **237**, 22-42 (2010); Levin-Klein R, Bergman Y. *Front Immunol* **5**, 625 (2014)]. We thank the reviewer for raising this question as we feel that the inclusion of a model that reiterates the mechanism leading to the imbalanced histone modifications between the IGH alleles would be a valuable addition to the manuscript. In the revision, we now include a new Fig. 5 which outlines the stepwise epigenetic process leading to allelic exclusion of IGH during B-cell development and projects the timing of DUX4 insertion based on translocation breakpoints, epigenetic profiling data and toxicity experiments. To provide further support for this model, we generated whole-genome bisulfite sequencing (WGBS) data for Nalm6. The result, presented in the newly generated Fig. 2c, shows hypermethylation in the IGH-DUX4 re-arranged haplotype, in contrast to the hypomethylation of the wild-type IGH allele, which is consistent with the model presented in Fig. 5. The model is presented in Discussion as follows:

“By integrating DNA translocation breakpoints, epigenetic profiling, transcription and cell toxicity data in Nalm6, we propose the following model for IGH-DUX4 translocation during B-cell development (Fig. 5) based on current knowledge of stepwise epigenetic process that controls the allelic exclusion of IGH^{13, 15, 18}. B-cell development initiates from hematopoietic stem cells with both IGH alleles hypo-acetylated and silenced by DNA hypermethylation^{13, 18}. This is followed by

the development of early progenitor B (pro-B) cells where both IGH alleles undergo D-to-J rearrangement. Structural variation (SV) breakpoints of IGH-DUX4 in Nalm6 occurred in the IGH D-J junction, suggesting that the IGH translocation arise at the pro-B cell stage as demonstrated previously by SV breakpoint analysis in IGH re-arranged multiple myeloma patient samples²³. Consistent with this, prior studies by our group and the others shown that the IGH-DUX4 fusions are clonal events acquired early in leukemogenesis^{4, 5}. At this stage, both IGH alleles remain methylated. In late pro-B and pre-B cell stage, one of the IGH alleles is randomly selected for activation by demethylation and hyperacetylation followed by the VDJ rearrangement. If an IGH-DUX4 rearranged allele were selected for activation (which is possible as not all B-ALLs express functional IGH^{24, 25}), the resulting highly expressed DUX4 would likely be too toxic to permit the survival of leukemia cells, leading to cell death (Fig. 4). By contrast, activation of the wild-type IGH allele would lead to moderate expression of DUX4 from the silenced IGH-DUX4 allele, ensuring leukemia cell viability.”

4. The author should provide evidence on the silenced IGH of IGH-DUX4 in the primary sample. For example, the author might perform the WGS/Hi-C/H3K27ac CHIP-seq/ATAC-seq in the primary sample with the IGH-DUX4 fusion and see if DUX4 was indeed fused to the silenced allele of IGH.

[Author Response] We agree with the reviewer that additional epigenetic profiling and haplotype analysis of primary patient samples could be performed, however given that most of these experiments require large number of cells (e.g. high resolution Hi-C needs ~1 million cells), xenograft mouse model will need to be established to provide sufficient material. This is unfortunately out of the scope of the current study. Therefore, in the revised manuscript, we emphasized the need for future investigation to generalize IGH-DUX4 to silenced allele in B-ALL. This is incorporated in the Discussion as follows:

*“Our analysis of RNA-seq data generated from 54 B-ALL pediatric patient samples with IGH@ translocation shown that SV breakpoints on IGH-DUX4 and IGH-CRLF2 translocation were highly enriched on IGH D-J junctions (red box on **Supplementary Fig. 6**) while expression of the target oncogenes, i.e. DUX4 (**Fig. 1**) and CRLF2 (**Supplementary Fig. 7**, data are from published paper^{17, 28}) is significantly lower than that of Igμ—both patterns match what we observed in Nalm6. The consistency between patient data and the Nalm6 cell line suggest that IGH@ proto-oncogene translocation on the silenced allele, discovered in Nalm6 through comprehensive analysis on haplotype structure and epigenetic profiling, could also occur in B-ALL patient samples. Further studies employing haplotype analysis and epigenomic profiling using patient samples or patient-derived xenograft mouse models are needed to provide additional support for this hypothesis.”*

Minor

1. Why are the SNPs used in Fig 1 and Fig 2 different?

[Author Response] The SNPs in Fig. 1 (Fig. 2 in revision version) were used to measure allele-specific expression at the IGHM locus. By contrast, those used for Fig. 2 (supplementary Fig. 5 in revision version) were to confirm that the interaction between enhancer E μ and I μ promoter is from the wild-type haplotype. We have revised the figures and clarified the selection of SNPs in the figure legend.

2. Fig 1a. The authors had better to compare the expression of IGH-fusion and wild-type DUXs, instead of wild-type IGH.

[Author Response] As stated in our response to major question #1, wild-type DUX4 is expressed only in embryo cell (shown as red dots in Fig. 1a) but not in normal pre-B or mature B cell. Oncogenic activation of DUX4 leading to its expression in the IGH-DUX4 subtype of B-ALL (shown as green dots for patients and blue dot for Nalm6 in Fig. 1a) was caused by hijacking IGH enhancer via translocation based on our prior study [Zhang J, et al. Nat Genet 48, 1481-1489 (2016)]. Comparison of IGH with DUX4 expression prompted us to investigate allele-specificity of DUX4 translocation in B-ALL and our final conclusion was based on extensive analysis of epigenetic profiling and haplotype analysis at the IGH loci where DUX4 translocation occurred.

3. DUX4 is mapped using hg38 while the expression DUX4 is measured using hg19. Please unify.

[Author Response] We used the reference sequence of DUX4L13, the DUX4 copy identified by local assembly of Chromium WGS at IGH locus as described in Methods. DUX4L13 on hg38 was used instead of hg19 to quantify DUX4 expression. In the revision, we compared the DUX4L13 sequence on hg19 and GRCh38 and found DUX4L13 is identical on these two assemblies (GRCh38 and hg19). To ensure consistency, we revised the method part “*DUX4L13 1,267 bp length, chr10: 133,753,250-133,754,516 in hg38*” to “*DUX4L13 1,267 bp length, chr10: 135,490,575-135,491,841 in hg19*”.

Reviewer #3 (Remarks to the Author):

The article entitled « Long-read sequencing unveils oncogenic translocation to silenced IGH allele in pediatric leukemia », by Tian et al. investigates the IGH-DUX4 translocation in a cell line by several novel technics (long read genome profiling, transcriptome sequencing, epigenetic and 3D profiling) and concludes that this translocation occurs on the silenced IGH allele.

Although the molecular data appear convincing, some of the conclusions are not supported by the data.

Why is it important to know that translocation can occur on the silenced copy? Are there medical or biological consequences and application to this observation? Only a weaker transactivation?

[Author Response] In the original manuscript, we described the biological consequence of DUX4 translocation to the silenced IGH allele was to overcome oncogenic stress of DUX4 based on prior studies that show apoptosis in rhabdomyosarcoma cell line TE671 with ectopic expression of DUX4. Our own apoptosis assay on murine hematopoietic cells shows the same pattern (results in the original Supplementary Fig. 4).

In the revised manuscript, by performing a new experiment on DUX4 oncogenic toxicity in Nalm6, we provide more support for this hypothesis and incorporated the findings into two new figures (Fig. 4 for experimental data, Fig. 5 for a new model for oncogene toxicity) to demonstrate that translocation at the silenced allele is likely to ensure viability of IGH-DUX4 leukemia cells by escaping oncogene “overdose”. In the revision, we incorporated the following statement in Discussion: *“Previously, Yasuda et al. shown that knockdown of the IGH-DUX4 fusion would suppress proliferation of Nalm6 cells⁴, confirming the oncogenic potential of DUX4 expression in B-ALL. This, coupled with the oncogenic toxicity caused by overexpression of DUX4 presented in our study (Fig. 4), suggests that DUX4 expression in Nalm6 may follow the “Goldilocks principle”²⁶ —i.e. “just-right” levels are required as too-much will lead to apoptosis while too-little will lead to suppression of proliferation. DUX4 translocation to the silenced allele may thus provide the selective advantage required to achieve the precise level of expression to promote fitness of leukemia cells. In this regard, targeting of DUX4 by considering the potential for exploiting its oncogenic stress^{26, 27} may provide a unique therapeutic angle for IGH-DUX4 B-ALL treatment.”*

As the haplotype and epigenetic analysis are based on Nalm6 alone, we also noted the limitation of the current study regarding potential medical application as outlined in our discussion as follows:

“Our analysis of RNA-seq data generated from 54 B-ALL pediatric patient samples with IGH@ translocation shown that SV breakpoints on IGH-DUX4 and IGH-CRLF2 translocation were

highly enriched on IGH D-J junctions (red box on **Supplementary Fig. 6**) while expression of the target oncogenes, i.e. DUX4 (**Fig. 1**) and CRLF2 (**Supplementary Fig. 7**, data are from published paper^{17, 28}) is significantly lower than that of Igh—both patterns match what we observed in Nalm6. The consistency between patient data and the Nalm6 cell line suggest that IGH@ proto-oncogene translocation on the silenced allele, discovered in Nalm6 through comprehensive analysis on haplotype structure and epigenetic profiling, could also occur in B-ALL patient samples. Further studies employing haplotype analysis and epigenomic profiling using patient samples or patient-derived xenograft mouse models are needed to provide additional support for this hypothesis.”

The authors use RNA-seq data to evaluate the expression level. RNA level reflect transcription but also RNA stability. To assure the comparison, could the author show that DUX4 RNA stability is comparable to the stability of the functional IGH RNA.

[Author Response] We appreciate the reviewer’s comment that many factors can contribute to the abundance of RNA expression. However, the focus of our investigation is on allelic-specificity of IGH-DUX4 translocation in the context of the reported allelic exclusion at this locus, rather than all potential factors that could alter expression level from RNA-seq. In this respect, the most relevant measurement is the expression specificity of the wild-type versus the re-arranged haplotype as presented in Fig. 2b. However, we agree that higher expression of IGH compared to DUX4 in patient samples should not be used as a direct evidence for IGH-DUX4 translocation into the silenced allele and have therefore modified the statement in the Abstract and Discussion to clarify this position as follows: “Patient samples of IGH-DUX4 B-ALL have similar expression profile and IGH breakpoints as Nalm6, suggesting this could be a common mechanism which requires further analysis of haplotype structure and epigenetic profiling” and “Further studies employing haplotype analysis and epigenomic profiling using patient samples or patient-derived xenograft mouse models are needed to provide additional support for this hypothesis.”

If one can agree that the translocated copy of the IGH locus is less expressed than the IGH copy carrying the functional rearrangement, this does not imply that the translocation occurred on a previously silenced copy. The translocation may occur on a copy and, at the end of the transformation process, the translocated locus ends up less transcribed than the one carrying the functional IGH rearrangement. So the authors do not provide any proof that the translocation occurs on a silenced copy. Furthermore, no reason is provided to extend the conclusion to other B-ALL translocations based only on the location of the breakpoints in the

D-J regions. Line 110-114: I do not understand the rationale: breakpoints at D and J regions are enriched for DUX4 and CRLF2, and so what ? please clarify

[Author Response] We thank the reviewer for this question. However we do believe there are multiple lines of evidence to suggest that the DUX4 translocation does in fact occur i) early in B-cell development, and ii) on a silenced copy of the allele. Given we have observed a DUX4 breakpoint site cluster between the D and J segments of IGH, this infers translocations occurred at a particular temporal stage of the IGH segment rearrangement process - early on in B-cell development at the early pro-B stage, during which each IGH loci were silenced (similar logic was presented in structural variant breakpoint analysis in IGH re-arranged multiple myeloma patients [Walker BA, et al. *Blood* **121**, 3413-3419 (2013)]). It is known that activation (demethylation) of one IGH allele subsequently occurs at a later stage (i.e. late pro-B/early pre-B). To confirm this in our study, in addition to showing lower expression of the translocation locus transcript, we have also shown in Fig. 2, in our evaluation of the “silenced IGH allele” in Nalm6 was indeed based on allelic imbalance of H3K27ac and ATAC-seq which measures enhancer activity of the translocated versus wild-type allele. In the revised manuscript, to provide further evidence on the epigenetic state of Nalm6, we performed whole-genome bisulfite sequencing (WGBS) which show hypermethylation at the translocated allele. The result is presented in the newly included Fig. 2c. Thus, given the observed breakpoint cluster, allelic imbalance of H3K27ac ChIP-seq, ATAC-seq, and WGBS results, we have provided experimental proof that the DUX4 translocation occurs on a silenced copy of the IGH locus. To clarify this position, we have now included a new Figure (Fig. 5) to illustrate the timing of translocation during B-cell development in the revision. Given the lack of haplotype structure and epigenetic/Hi-C data in patient samples, we have also re-iterated in the revised manuscript that translocation to the silenced allele in patient sample is a hypothesis that requires further investigation. These points are included in the Discussion as follows:

“By integrating DNA translocation breakpoints, epigenetic profiling, transcription and cell toxicity data in Nalm6, we propose the following model for IGH-DUX4 translocation during B-cell development (Fig. 5) based on current knowledge of stepwise epigenetic process that controls the allelic exclusion of IGH^{13, 15, 18}. B-cell development initiates from hematopoietic stem cells with both IGH alleles hypo-acetylated and silenced by DNA hypermethylation^{13, 18}. This is followed by the development of early progenitor B (pro-B) cells where both IGH alleles undergo D-to-J rearrangement. Structural variation (SV) breakpoints of IGH-DUX4 in Nalm6 occurred in the IGH D-J junction, suggesting that the IGH translocation arise at the pro-B cell stage as demonstrated previously by SV breakpoint analysis in IGH re-arranged multiple myeloma patient samples²³. Consistent with this, prior studies by our group and the others shown that the IGH-DUX4 fusions are clonal events acquired early in leukemogenesis^{4, 5}. At this stage, both IGH alleles remain methylated. In late pro-B and pre-B cell stage, one of the IGH alleles is randomly selected for activation by demethylation and hyperacetylation followed by the VDJ rearrangement. If an IGH-

DUX4 rearranged allele were selected for activation—which is possible as not all B-ALLs express functional IGH^{24, 25}, the resulting highly expressed DUX4 would likely be too toxic to permit the survival of leukemia cells, leading to cell death (Fig. 4). By contrast, activation of the wild-type IGH allele would lead to moderate expression of DUX4 from the silenced IGH-DUX4 allele, ensuring leukemia cell viability.

Previously, Yasuda et al. has shown that knockdown of the IGH-DUX4 fusion would suppress proliferation of Nalm6 cells⁴, confirming the oncogenic potential of DUX4 expression in B-ALL. This, coupled with the oncogenic toxicity caused by overexpression of DUX4 presented in this study (Fig. 4), suggests that DUX4 expression in Nalm6 may follow the “Goldilocks principle”²⁶ — i.e. “just-right” levels are required as too-much will lead to apoptosis while too-little will lead to suppression of proliferation. DUX4 translocation to the silenced allele may thus provide the selective advantage required to achieve the precise level of expression to promote fitness of leukemia cells. In this regard, targeting of DUX4 by considering the potential for exploiting its oncogenic stress^{26, 27} may provide a unique therapeutic angle for IGH-DUX4 B-ALL treatment.

*Our analysis of RNA-seq data generated from 54 B-ALL pediatric patient samples with IGH@ translocation shown that SV breakpoints on IGH-DUX4 and IGH-CRLF2 translocation were highly enriched on IGH D-J junctions (red box on **Supplementary Fig. 6**) while expression of the target oncogenes, i.e. DUX4 (**Fig. 1**) and CRLF2 (**Supplementary Fig. 7**, data are from published paper^{17, 28}) is significantly lower than that of Igh μ —both patterns match what we observed in Nalm6. The consistency between patient data and the Nalm6 cell line suggest that IGH@ proto-oncogene translocation on the silenced allele, discovered in Nalm6 through comprehensive analysis on haplotype structure and epigenetic profiling, could also occur in B-ALL patient samples. Further studies employing haplotype analysis and epigenomic profiling using patient samples or patient-derived xenograft mouse models are needed to provide additional support for this hypothesis.”*

What is the range and diversity of IGH copy expression in B-ALLs? Is the expression ratio between functional/nonfunctional copy different between IGH-translocated and non translocated samples? Is there any hint on the differentiation steps those samples are arrested at, with respect to normal B-cell development?

[Author Response] Igh μ expression in B-ALLs with IGH@ translocation (i.e. IGH-CRLF2, IGH-DUX4 and IGH-EPOR) is comparable to those without IGH@ translocation based on our analysis of Igh μ expression level using RNA-seq data (297 samples from different subtypes) of St. Jude B-ALL patients (see figure below). We can not discriminate the expression from functional versus non-functional IGH allele in these samples due to lack of long-read sequencing data, H3K27ac ChIP-

seq or DNA methylation data in these samples. We have stated this limitation in our analysis of B-ALL patient samples of IGH@ translocation in the Discussion.

SupFig4 : The used vectors need to be detailed, and the levels of expression of the DUX4 proteins could be checked. What are the transduced cells? Bone marrow cells are a mix of cells. Do they represent the cells in which the translocation occurred? Or at minimal, do they represent the tumor cells?

[Author Response] The reviewer raised a very important question regarding the cell types used for the DUX4 transfection experiment. Bone marrow cells are indeed comprised of different cell types, the majority of which are immature T and B cells but not the pre-B cells of IGH-DUX4 B-

ALL subtype. Therefore, we conducted a new toxicity experiment by transfecting a FLAG-tagged DUX4 gene into the Nalm6 B-ALL cell line that already carries the IGH-DUX4 translocation. The results, presented along with the murine bone marrow study (originally shown on SupFig4), are shown in a newly included Figure 4 (panel C and D); details on the murine bone marrow and Nalm6 experiments including the cloning vectors used, are described in the revised Methods section of “Apoptosis Assay”. In the revision, we incorporated the following statement in Results:

“To further evaluate the effects of DUX4 overexpression in fully transformed leukemia cells, we introduced DUX4 into Nalm6 cells which already harbor the IGH-DUX4 translocation, confirming GFP-tagged DUX4 protein expression by Western blot (Fig. 4c and 4d). The results shown that fully transformed leukemia cells would not tolerate further overexpression of DUX4 with a significant increase in apoptosis when comparing DUX4 transduced cells with the empty vector control ($p=0.0124$). We therefore conclude that while low levels of the DUX4 protein are tolerable, high levels induce apoptosis providing a biologic rationale for expression from the repressive haplotype.”

Are the Nalm6 cells diploid? Do they have only one copy of each IGH locus?

[Author Response] Nalm6 is “near diploid” as described in the cytogenetic section of HyperCLDB database (<http://bioinformatics.hsanmartino.it/cldb/cl3632.html>). We added this information in Introduction.

The IGH locus in Nalm6 has undergone VDJ recombination in Nalm6 as expected for pre-B cells. The coverage of chromium WGS shown V/D/J deletion spanning the majority of IGH region while the region spanning IGHM and E μ is diploid (Fig. 3a).

Minor remarks

Line 34: between

[Author Response] We thank the reviewer for pointing out this typo and have made the correction.

Line 37: please precise where does the DUX4 gene lies ? : chromosome 4 (4q35 ?), chromosome 10 ? where ?

[Author Response] We have included the details in Introduction: “DUX4 is located within the GC-rich D4Z4 repeat array at the subtelomeric regions of 4q35 and 10q26 which are characterized by high levels of repression^{7, 8}.”

Line 46 : please explain what means FPKM

[Author Response] We added the following in the Introduction: “*FPKM (Fragments Per Kilobase of transcript, per Million mapped reads)*”.

Line 112 : how many patients

[Author Response] We used all 54 B-ALL patient samples with IGH@ translocations from St. Jude Pediatric Cancer (PeCan) data portal and revised the manuscript by including the patient number in Discussion and in the legend of supplementary Fig. 6 as follows:

*“Our analysis of RNA-seq data generated from 54 B-ALL pediatric patient samples with IGH@ translocation shown that SV breakpoints on IGH-DUX4 and IGH-CRLF2 translocation were highly enriched on IGH D-J junctions (red box on **Supplementary Fig. 6**)”*

“54 B-ALL samples with IGH @ translocations were shown, which were obtained from St. Jude Pediatric Cancer (PeCan) data portal².”

Figure1. « only exonic SNPs » : please precise which ones

[Author Response] We have added the information in the revised Fig. 2 legend: “*For RNA-seq, read-out is available only for the two exonic SNPs (rs1059713, rs1136534)*”.

Figure 2a and sup4 : please explain why chromosome 10, and not chromosome 4.

[Author Response] Local assembly of IGH locus using chromium WGS shown that DUX4L13, which is located on chromosome 10, was translocated to IGH locus in Nalm6. The assembled contig shown 100.0% identity (no mismatch out of 1267bp) to DUX4L13 on chromosome 10 but the best match on chromosome 4 is 99.67% (4 mismatch out of 1218bp) at the DUX4L4 locus. Its sequence identity to canonical DUX4 (Gencode v28lift37) gene on chromosome 4 is 99.29% (3 mismatch and 1 gap of 6bp). RNA-seq reads from Nalm6 are mapped with 100% identity to the assembled contig, supporting DUX4L13 locus was translocated and expressed (**Supplementary Fig. 2b**). This was further corroborated by much broader interaction of IGH and DUX4 region on chromosome 10 (**Supplementary Fig. 4b**) compared to that on chromosome 4 (**Supplementary Fig 4a**) from Hi-C data in Nalm6.

We added the following Result: “*The translocated DUX4 sequence shares 100% sequence identity to DUX4L13 located on chromosome 10 while its best match on chromosome 4 is the DUX4L4 locus with 99.67% (4 mismatches) sequence identity. RNA-seq reads share 100% identity to the assembled DUX4 contig, supporting exclusive transcription of DUX4L13 activated by translocation from chromosome 10 to IGH in Nalm6 (Supplementary Fig. 2b). Therefore, IGH-DUX4 translocation in Nalm6 occurred between chromosome 10 and 14.*”

“The Hi-C data demonstrates that the interaction of IGH locus on chromosome 14 with the DUX4 region on 10q26 is much broader than that with the DUX4 region on 4q35 (Supplementary Fig. 4), consistent with our finding that the translocated DUX4 was from chromosome 10 based on the assembled IGH-DUX4 haplotype and expressed transcripts by RNA-seq and Iso-Seq.”

Please explain what means ACTB

[Author Response] We added the explanation of ACTB in revised Figure 4 legend: *“beta actin (ACTB)”*.

Reviewer #4 (Remarks to the Author):

In this study an extensive range of state-of-the-art technologies is used to analyze the haplotypes involved in the IgH-DUX4 translocation present in the Nalm6 B-ALL cell line. The authors clearly demonstrate that it is the silenced IgH allele that is fused to the DUX4 locus, which suggests that DUX4 upregulation in the Nalm6 cell line is the result of hijacking the E μ enhancer on the silenced IgH allele. In an attempt to extend this observation made in a single cell line, cells from B-ALL patients carrying an IgH@ translocation are taken and analyzed for the expression levels of the translocation partner genes. These are lower than the levels observed for Ig μ in these patients, which is interpreted as support for a common mechanism in B-ALL in which the silenced IgH allele is selected for translocations.

Except for some minor comments below, I find this a technically well performed study. However, for reasons clarified below, I have doubts about the biological significance of these findings. I also have concerns about the authors' interpretation of differences in expression levels.

[Author Response] We thanked the reviewer for his/her positive comments on the extensive analysis that we performed on the Nalm6 cell line. Regarding reviewer's concern on biological insight, we believe that the new experiment on DUX4 toxicity assay in the Nalm6 cell line included in the revision (Fig. 4) provided strong evidence that oncogenic toxicity of DUX4 could lead to selection of IGH-DUX4 translation to the silenced IGH allele. In the revision, we have outlined this model in the newly included Fig. 5 based on the analysis of epigenetic data and structural variation breakpoint on IGH; details are included in our response to question #1 from the reviewer.

Based on the reviewer's feedback, we have also toned down the generalization of this mechanism in patient samples throughout the manuscript as more analysis on haplotype structure and epigenetic profiling is required to reach the conclusion despite the consistency between Nalm6 and patient samples in their gene expression level and location of IGH SV breakpoint. Details are described in our response to questions #2 and #11.

1. I am not an expert in B cell development, but is it possible that in order to become a B-cell with acute lymphoblastic leukemia (B-ALL) properties, successful rearrangement and production of IgH (Ig μ in case of the Nalm6 B-ALL cell line used here) is a prerequisite, otherwise you are not a B-cell? If so, to me this would imply that in B-ALL it will always be only the non-productively rearranged (silenced) IgH allele that is available for rearrangements to an oncogene and that the findings presented here would be entirely unsurprising? Authors, please discuss this issue.

[Author Response] We would like to clarify that 1) Functional IGH is NOT always required for B-ALL. A study by Trageser et al [J Exp Med 206, 1739-1753 (2009)] shown that 47 of 57 (83%) of patient-derived Ph+ (Philadelphia chromosome) B-ALL cases carried only nonfunctional *IGHM* V_HDJ_H gene rearrangements. More recently, Geng et al. found that only 13.5% of the 830 B-ALL cases they analyzed had functional pre-BCR (B-cell receptor, a complex including IGH protein), an important checkpoint during early B-cell development [Geng H, et al. Cancer Cell 27, 409-425 (2015)]. 2) IGH translocation occurred at the early progenitor B-cell stage (pro-B) when both IGH alleles are still methylated, so both alleles are available for the subsequent VDJ rearrangements which occurs at the late pro-B or pre-B stage.

In the revised manuscript, we included a new figure (Fig. 5) to illustrate IGH rearrangement and IGH-DUX4 translocation in the context of B-cell development and to clarify these two points raised by the reviewer in the section of Discussion as follows:

“By integrating DNA translocation breakpoints, epigenetic profiling, transcription and cell toxicity data in Nalm6, we propose the following model for IGH-DUX4 translocation during B-cell development (Fig. 5) based on current knowledge of stepwise epigenetic process that controls the allelic exclusion of IGH^{13, 15, 18}. B-cell development initiates from hematopoietic stem cells with both IGH alleles hypo-acetylated and silenced by DNA hypermethylation^{13, 18}. This is followed by the development of early progenitor B (pro-B) cells where both IGH alleles undergo D-to-J rearrangement. Structural variation (SV) breakpoints of IGH-DUX4 in Nalm6 occurred in the IGH D-J junction, suggesting that the IGH translocation arise at the pro-B cell stage as demonstrated previously by SV breakpoint analysis in IGH re-arranged multiple myeloma patient samples²³. Consistent with this, prior studies by our group and the others shown that the IGH-DUX4 fusions are clonal events acquired early in leukemogenesis^{4, 5}. At this stage, both IGH alleles remain methylated. In late pro-B and pre-B cell stage, one of the IGH alleles is randomly selected for activation by demethylation and hyperacetylation followed by the VDJ rearrangement. If an IGH-DUX4 rearranged allele were selected for activation—which is possible as not all B-ALLs express functional IGH^{24, 25}, the resulting highly expressed DUX4 would likely be too toxic to permit the survival of leukemia cells, leading to cell death (Fig. 4). By contrast, activation of the wild-type IGH allele would lead to moderate expression of DUX4 from the silenced IGH-DUX4 allele, ensuring leukemia cell viability.”

2. The authors state in their introduction “Intriguingly, DUX4 expression was much lower than Igm in all 32 patients examined (the median FPKM of DUX4 is only 21% of that of Igm, Fig. 1a), raising the alternative possibility that IGH-DUX4 translocation might occur on the silenced IGH allele”. Later in the text (line 114) they again compare expression levels of Igm to that of translocation partner genes, to draw a similar conclusion: In IGH-DUX4 patients, we found

lower expression of DUX4 than Ig μ (Fig. 1a). The same pattern was also found in IGH-CRLF2 patients— the median FPKM of CRLF2 is only 26% of that of Ig μ from published RNA-seq data of 24 IGH-CRLF2 patients^{12, 17} (Supplementary Fig. 7). This suggests that oncogenic translocation to the silenced IGH allele could be a common mechanism in B-ALL.”

In my opinion, this data cannot be presented as indicative for it being the enhancer on the silenced IgH allele that is responsible for transcription of the translocated gene. After all, expression levels not only depend on enhancer strength but also on the enhancer-promoter combination (we do not know DUX4 levels under the control of the active IgH enhancer: even at its full power, it may still not be able to activate the DUX4 promoter to Ig μ levels!), enhancer-promoter distance, gene length, mRNA stability, etc, etc.

[Author Response] We thank the reviewer for this important point. As evident in our original manuscript, we believe we have provided strong evidence to indicate that the IGH enhancer on the silenced allele is responsible for transcription of the translocated DUX4 in the Nalm6 cell line. This evidence was derived from the extensive long-read sequencing and epigenetic profiling performed in Nalm6. However, we entirely agree with the reviewer that the patient data presented, involving Igu and DUX4 or CRLF2 expression data and also conserved IGH breakpoint patterning, does not provide direct evidence that the same mechanism occurs in patients. To generate extensive long-read sequencing and epigenetic profiling for patients will require establishing patient-derived xenograft mouse model to harvest the large number of cells required for these experiments. In any case, we do want to emphasize that translocation to the silenced IGH allele remains a valid hypothesis in the patient samples as the clusters of translocation breakpoints and expression level in patient samples match the model that we proposed based on the Nalm6 data presented in the new Fig. 5. The lower DUX4 expression compared to IGH in patient samples, consistent with what we observed in Nalm6, also suggest that the hypothesis of translocation to the silenced allele is plausible. Therefore, we included the following statement in the revised manuscript in the section of Discussion: *“The consistency between patient data and the Nalm6 cell line suggest that IGH@ proto-oncogene translocation on the silenced allele, discovered in Nalm6 through comprehensive analysis on haplotype structure and epigenetic profiling, could also occur in B-ALL patient samples. Further studies employing haplotype analysis and epigenomic profiling using patient samples or patient-derived xenograft mouse models are needed to provide additional support for this hypothesis.”*

Line 83: To quantify the contact intensity between E μ -DUX4 and the E μ -Ig μ promoter, we designed Capture-C with capture probes in E μ , and found that the contact intensity of E μ -DUX4 was ~14-fold lower than that of the E μ -Ig μ promoter (Fig. 2a). Therefore, the lower

expression of DUX4 compared to Ig μ (~15-fold, blue dot in Fig. 1a) likely results from the weaker enhancer-promoter interaction of E μ -DUX4 compared to E μ -Ig μ promoter.

This statement and the analyses raises several questions:

3. It is unclear from Figure 2A where the DUX4 gene starts and ends, i.e. where to look for enhancer-promoter interactions: authors, please properly annotate plots for readers to orient themselves.

[Author Response] We thank the reviewer for this great suggestion. In the revised manuscript we modified Fig. 3 (the original Fig. 2) by adding a gene track on Fig. 3A to annotate the location of DUX4 genes highlighting that two copies of DUX4 were involved in translocation. Supplementary Fig. 3 which shows the HiChIP data was also modified similarly. Recognizing that enhancer-promoter structure on the re-arranged IGH-DUX4 allele is different from the wild-type allele (aka the reference genome), we included a new panel (Fig. 3b) depicting the gene structures at the two IGH alleles. This should provide additional “orientation” for evaluating the E μ interaction with DUX4 versus Ig μ . The revision also modified y-axis scale for the two interactions in Fig. 3 and Supplementary Fig. 3 based on the reviewer’s question #9.

4. As the authors acknowledge themselves: many copies of DUX4 exist in the reference genome, making it very difficult to assign mapped reads to a given DUX4 copy. To me this seems to compromise proper quantification of contact frequencies. Authors, please clarify this issue and justify your quantification strategy in the main text.

[Author Response] We thank the reviewer for this great question. Local assembly of IGH-DUX4 haplotype derived from the Chromium WGS reads mapped to the *IGH* region (described in the Methods section) shown that DUX4L13 on chromosome 10 is the best match to the translocated DUX4 (exact the same as our assembled sequence), thus we used DUX4L13 as the template to map RNA-seq reads for quantification as described in the original manuscript. RNA-seq reads at DUX4 loci are mapped with 100% identity to the assembled DUX4 copies, supporting DUX4L13 locus was translocated and expressed (**Supplementary Fig. 2b**). This was further corroborated by much broader interaction of IGH and DUX4 nearby region on chromosome 10 (Supplementary Fig. 4b) compared to that on chromosome 4 (Supplementary Fig 4a) from Hi-C data in Nalm6.

We used the highest peak for quantification of contact frequencies in Capture-C presented in Fig. 3C. Among all DUX4 clusters based on the annotation of Gencode V28lift37, only one peak at DUX4L13 was detected with a coverage of 56 reads, matching the contig derived from local assembly of IGH locus (shown in Fig. 3a). Genome-wide, a total of 101 reads were mapped to any DUX4 clusters. To address the reviewer’s concern, we mapped all 101

reads to the DUX4L13 region (chr10:135,488,523-135,493,885 on hg19) and found that the highest peak has a sequence coverage of 75. To account for the multiple mapping issue, we used 75 as the peak height for DUX4 and readjusted the fold change of interaction intensity between E μ -I γ μ promoter and E μ -DUX4 accordingly in the revision.

As described in our response to the reviewer's question #6, we used the similar approach to account for multiple mapping issue at DUX4 clusters to quantify E μ -DUX4 interaction in Hi-C and HiChIP data.

To clarify the multiple mapping issues, we revised Fig. 3 legend as follows: *“(a) The two translocated DUX4 copies were marked at DUX4L13 region, because this region is best matched to our assembled IGH-DUX4 haplotype. Details are in Methods. (c) Comparison of supporting read pairs for Hi-C/H3K27ac HiChIP, the reads coverage for Capture-C between E μ -DUX4 interaction and E μ -I γ μ promoter interaction. Multiple mapping issue at DUX4 regions was considered. The details are described in Methods.”*

In Results: *“The translocated DUX4 sequence shares 100% sequence identity to DUX4L13 located on chromosome 10 while its best match on chromosome 4 is the DUX4L4 locus with 99.67% (4 mismatches) sequence identity. RNA-seq reads share 100% identity to the assembled DUX4 contig, supporting exclusive transcription of DUX4L13 activated by translocation from chromosome 10 to IGH in Nalm6 (**Supplementary Fig. 2b**). Therefore, IGH-DUX4 translocation in Nalm6 occurred between chromosome 10 and 14.”*

*“The Hi-C data demonstrates that the interaction of IGH locus on chromosome 14 with the DUX4 region on 10q26 is much broader than that with the DUX4 region on 4q35 (**Supplementary Fig. 4**), consistent with our finding that the translocated DUX4 was from chromosome 10 based on the assembled IGH-DUX4 haplotype and expressed transcripts by RNA-seq and Iso-Seq.”*

We also revised Methods by incorporating the following text to address the multiple mapping issues:

“Our local assembly of IGH-DUX4 haplotype derived from the Chromium WGS shown DUX4L13 on the reference genome is the best match to the translocated DUX4.”

“We mapped the ~23 kb sequence to hg19 and found the assembled DUX4 region was best matched to DUX4L13. DUX4L13 shown exact the same as our assembled sequence. The other DUX4 copies in hg19 shown at least 1 mismatch with the assembled sequence.”

*“The highest peak at DUX4 region and I γ μ promoter region were used to quantify E μ -DUX4 and the E μ -I γ μ interaction. Although many DUX4 copies exist in human genome, only one peak of 56 reads at DUX4L13 (the exact same sequence as the DUX4 sequence on our assembled IGH-DUX4 haplotype in Nalm6) was found (shown in **Fig. 3a**). The enrichment of Capture-C mapping to DUX4L13 can be explained by its ≥ 1 -bp difference with all other annotated DUX4 genes on hg19. To account for multiple mapping issue, we identified a total of 101 reads mapped to any DUX4 clusters based on the annotation of Gencode V28lift37 and mapped these 101 reads*

to the DUX4L13 nearby region (chr10:135,488,523-135,493,885 in hg19) using *bwa*⁴⁴, resulting in an increase of DUX4L13 peak coverage to 75. Therefore, the fold change of interaction intensity between E μ -DUX4 and the E μ -I μ promoter was estimated to be 13 fold (953 vs. 75)."

5. Contacts with DUX4 are less frequent than with the I μ promoter. What is being compared here (what do the boxes represent): contacts across the entire DUX gene versus contacts with the I μ promoter? Please specify and justify the regions that are compared here.

[Author Response] Comparison of contact intensity of E μ -DUX4 and E μ -I μ promoter was based on Capture-C peaks present at DUX4 and I μ . E μ -DUX4 interaction was derived from the single-peak mapped to DUX4 within the exonic region of DUX4L13 (the sequence is exactly the same as our assembled sequence of the IGH-DUX4 haplotype from Chromium WGS data) and the peak location is likely to be affected by the short gene size of DUX4, a single-exon gene of 1,267bp (DUX4L13 in hg19) as well as the imprecision between Capture-C peak and the real interaction locus [Davies JO, et al. Nat Methods 13, 74-80 (2016)]. Similarly, E μ -I μ promoter interaction is based on the peak present at I μ located at the 5' UTR. We clarified this in the Methods along with details on addressing multiple mapping issues on DUX4 in our response to question #4.

In the revised manuscript, we also added a new panel Fig. 3c to present quantification of Capture-C data along with Hi-C and Hi-ChIP as requested by reviewer's next question and the details of the quantification are described in Methods as documented in our response to question #4.

6. The authors use Capture-C data to quantify contact frequencies but they could also do this with the Hi-C data (by windowing the data). Please provide this analysis as well.

[Author Response] We thank the reviewer for this great suggestion. For Hi-C and Hi-ChIP data, to quantify the contact frequency, we first collected the list of paired alignments from the Hi-C or Hi-ChIP pipeline result, then counted the read pairs with one read mapped to E μ and the other mapped to DUX4 or I μ promoter. This ensures that the measurement only includes those that are involved in E μ -DUX4 contact or E μ -I μ promoter contact. E μ enhancer (1859 bp) and I μ promoter (4004 bp), as defined in Supplementary Fig. 8a and 8b, were based on H3K27ac/ATAC-seq signals. We included all regions of DUX4 cluster based on the annotation of Gencode V28lift37 to define interaction of E μ -DUX4, i.e. a read pair with one read falls in E μ and the other in any DUX4 clusters. For any qualified DUX4 read-pair, we extracted the DUX4 read and confirmed its identity to DUX4L13 by mapping the read to the DUX4L13 region as specified in our response to reviewer's question #4. In the revised manuscript, quantification

based on read-pair analysis of Hi-C and Hi-ChIP are presented on Fig. 3c, confirming the result based on peak analysis of Capture-C. The details of the data analysis are presented in Methods:

“Only the read pairs that one read falls in $E\mu$ and the other in the $Ig\mu$ promoter were used to evaluate the $E\mu$ - $Ig\mu$ promoter interaction. Because of multiple mapping issue for DUX4 region, to quantify $E\mu$ -DUX4 interaction reads, all the read pairs that one read falls in $E\mu$ and the other in any DUX4 array region annotated by Gencode V28lift37 were defined as the read pairs supported $E\mu$ -DUX4 interaction.” “The same defined enhancer/promoter region and the same strategy for Hi-C data analysis were used to quantify $E\mu$ -DUX4 interaction reads and $E\mu$ - $Ig\mu$ promoter interaction reads in H3K27ac HiChIP data.”

We incorporated the following text in Result: *“ $E\mu$ - $Ig\mu$ promoter interaction measured by read-pair count in Hi-C and H3K27ac HiChIP is 3 and 8 fold of that of the $E\mu$ -DUX4 interaction, respectively (Fig. 3c, details in Methods). The stronger $E\mu$ and $Ig\mu$ interaction is not related to the distance-associated chromatin interaction decay¹⁹ because the distance between $E\mu$ and $Ig\mu$ promoter on the wild-type haplotype (~140Kb) is in fact much longer than that (~4Kb) between $E\mu$ and DUX4 on the IGH-DUX4 haplotype (Fig. 3b, Methods). To verify this, we designed Capture-C with “bait” probes around $E\mu$ and performed coverage-based peak analysis, which shows that $E\mu$ - $Ig\mu$ promoter interaction is 13 fold higher than that of $E\mu$ -DUX4 interaction (Fig. 3c, Methods). The uniform pattern of weaker $E\mu$ -DUX4 interaction emerging from Hi-C, HiChIP, and Capture-C data provides further support that IGH-DUX4 haplotype was epigenetically silenced, consistent with allele-specific hypermethylation of this haplotype (Fig. 2c).”*

7. Contact frequencies with DUX4 are lower: is this because of differences in the linear distances between the $E\mu$ - $Ig\mu$ promoter and the $E\mu$ -DUX4 promoter? Please specify these distances and discuss if there are differences.

[Author Response] We thank the reviewer for raising this important question. In fact, the linear distance between $E\mu$ and DUX4 is ~4 kb on the re-arranged haplotype, much shorter than the ~140 kb linear distance between $E\mu$ and $Ig\mu$ promoter on the other haplotype (shown in new Fig. 3b). Therefore, the lower contact frequency with DUX4 is not a result of distance-dependent decay of chromatin interaction [Lajoie BR, et al. Methods. 2015 Jan 15;72:65-75].

The Hi-C data presented on Fig. 3 was mapped to the reference genome and does not scale proportionally because the region on chromosome 10 is much shorter than the region on chromosome 14. To clarify this point, we made the following two modifications. First, we labeled the genomic-scale on the chromosome 10 and chromosome 14 reference genome in panel A. Second, we generated a new panel B in Fig. 3 which specifically shows the wild-type haplotype and DUX4 re-arranged haplotype with the genomic distance between $E\mu$ - $Ig\mu$

promoter and the E μ -DUX4 labeled. Construction of the haplotypes takes into account the VDJ re-arrangement leading to the loss of IGH V region on one haplotype. Details are described in Methods “*Genomic distance between E μ and I μ promoter/DUX4*”.

We incorporated the following text in the Results: “*E μ -I μ promoter interaction measured by read-pair count in Hi-C and H3K27ac HiChIP is 3 and 8 fold of that of the E μ -DUX4 interaction, respectively (Fig. 3c, details in Methods). The stronger E μ and I μ interaction is not related to the distance-associated chromatin interaction decay¹⁹ because the distance between E μ and I μ promoter on the wild-type haplotype (~140Kb) is in fact much longer than that (~4Kb) between E μ and DUX4 on the IGH-DUX4 haplotype (Fig. 3b, Methods). To verify this, we designed Capture-C with “bait” probes around E μ and performed coverage-based peak analysis, which shows that E μ -I μ promoter interaction is 13 fold higher than that of E μ -DUX4 interaction (Fig. 3c, Methods). The uniform pattern of weaker E μ -DUX4 interaction emerging from Hi-C, HiChIP, and Capture-C data provides further support that IGH-DUX4 haplotype was epigenetically silenced, consistent with allele-specific hypermethylation of this haplotype (Fig. 2c).”*

8. The Hi-C data in Figure 2A appear to give lower resolution contact maps at the DUX4 region than at the IgH locus: how is this possible?

[Author Response] The resolution (5kb) of the Hi-C contact map is the same at DUX4 and IGH locus in Fig. 3A (the original Fig. 2A). However, the genomic span of the DUX4 region on chromosome 10 was 72Kb (left), much shorter than the 1.4Mb region of the IgH locus on chromosome 14 (right), which makes the Hi-C data contact map at the DUX4 region appear to be at a lower resolution. The different scale used in Fig. 3A enables a precise annotation of the DUX4 cluster which spans 20Kb while depicting all the main interactions with IGH enhancer (E μ), including E μ -I μ promoter, E μ -IGHA, E μ -IGHE and the E μ -DUX4 across the ~1.3Mb IgH locus.

In the revision, we added genomic scale to Fig. 3A to highlight the differences at the DUX4 (left) and the IGH (right) loci. In addition, we also included a new Supplementary Fig. 5a to show the Hi-C interaction plot on the scale for the 1.4 Mb region of chr10 subtelomere and the IGH region.

9. Please adjust the Capture-C scaling at the DUX4 region (is now 0-500, make it e.g. 0-20). One should see where the rearranged part of the chromosome starts and be able to judge the signal at the DUX4 promoter in the context of that of its immediate surrounding.

[Author Response] We thank the reviewer for this suggestion. In the revised Fig. 3a, different scale is used for Capture-C at DUX4 (scaled to 0-60) and IGH region (scaled to 0-1000). To ensure consistency, we also rescaled RNA-seq, H3K27ac and ATAC-seq tracks. These changes

were described in figure legend. The same changes were also made for supplementary Hi-C and HiChIP figures (supplementary Fig. 3, 4, 5).

We marked the location of translocated DUX4 in Fig. 3A as well as in supplementary Fig. 3 (the H3K27ac HiChIP plot). A schematic drawing of the translocated haplotype is presented in Fig. 3b to give further resolution.

As described previously, contact signals in other DUX4 copies could be caused by DUX4 mapping ambiguity. As described in our response to question #5, despite the presence of multiple DUX4 copies in the genome, a single Capture-C peak was detected in the exonic region of DUX4L13 (the sequence is exactly the same as our assembled sequence of the IGH-DUX4 haplotype from Chromium WGS data) representing the interaction of DUX4 with E μ enhancer as this peak location is likely to be affected by the short gene size of DUX4, a single-exon gene of 1,267bp (DUX4L13 in hg19) as well as the imprecision between Capture-C peak and the real interaction locus [Davies JO, et al. Nat Methods 13, 74-80 (2016)].

10. The author state: ‘We hypothesize that the IGH-DUX4 translocation is exclusively expressed on the silenced haplotype because higher expression levels of DUX4 from an active E μ cannot be tolerated without triggering apoptosis’. Please see my first comment: if it is true that in B-ALL only the silenced allele is available for translocations, all such speculations seem irrelevant.

[Author Response] As detailed in our response to question #1, *IGH-DUX4* translocation in Nalm6 is likely to arise at the progenitor B cell stage when both IGH alleles are silent based on our analysis of the structural variation breakpoints.

11. For reasons explained in comment 2, I find the observation that expression levels of translocation partner genes are lower than that of Ig μ in B-ALL patients carrying an IGH@ too circumstantial to be presented as support for translocations with the silenced IgH allele. In my opinion, authors should limit their conclusions throughout the text (including title, abstract) to their carefully analyzed Nalm6 cell line. Alternatively, they should haplotype the translocated chromosomes across a larger panel of B-ALL patients/cell lines.

[Author Response] We thank the reviewer for this suggestion, and we have now limited our conclusion in both abstract and discussion as follows. In the abstract, we now state “*Patient samples of IGH-DUX4 B-ALL have similar expression profile and IGH breakpoints as Nalm6, suggesting this could be a common mechanism which requires further analysis of haplotype structure and epigenetic profiling*”.

We incorporated the following text in the discussion. “*Our analysis of RNA-seq data generated from 54 B-ALL pediatric patient samples with IGH@ translocation shown that SV breakpoints on IGH-DUX4 and IGH-CRLF2 translocation were highly enriched on IGH D-J junctions*”

*(red box on **Supplementary Fig. 6**) while expression of the target oncogenes, i.e. DUX4 (**Fig. 1**) and CRLF2 (**Supplementary Fig. 7**, data are from published paper^{17, 28}) is significantly lower than that of Igμ—both patterns match what we observed in Nalm6. The consistency between patient data and the Nalm6 cell line suggest that IGH@ proto-oncogene translocation on the silenced allele, discovered in Nalm6 through comprehensive analysis on haplotype structure and epigenetic profiling, could also occur in B-ALL patient samples. Further studies employing haplotype analysis and epigenomic profiling using patient samples or patient-derived xenograft mouse models are needed to provide additional support for this hypothesis.”*

REVIEWERS' COMMENTS:

Reviewer #1 (Remarks to the Author):

The authors have sufficiently addressed my concerns in their resubmission.

Reviewer #2 (Remarks to the Author):

The authors made a significant improvement to address my comments. I think there are very few remaining points that have to be addressed:

The authors claimed that a "just-right" levels are required for ALL as too much will lead to apoptosis and too-little will lead to suppression of proliferation. However, as shown in Figure 1A, the FPKM of DUX4 ranges from 30 to 500 in patients with IGH-DUX4, which seems not so "just-right" and might conflict with the so-called "Goldilocks principle".

Allele specific knockdown experiments may provide the direct evidence for the IGH-DUX4 translocation occurs on the silenced IGH allele, although it is certainly difficult to develop suitable sgRNAs that target the individual allele.

Reviewer #3 (Remarks to the Author):

minor remarks

some typos?

eg have shown or showed instead of shown?

Reviewer #4 (Remarks to the Author):

The authors have carefully addressed my concerns, implemented important additional controls and toned down their conclusions, resulting in a much improved manuscript.

REVIEWERS' COMMENTS:

Reviewer #1 (Remarks to the Author):

The authors have sufficiently addressed my concerns in their resubmission.

[Author Response] We thank the reviewer for helping us improve the quality of the manuscript.

Reviewer #2 (Remarks to the Author):

The authors made a significant improvement to address my comments. I think there are very few remaining points that have to be addressed:

1 The authors claimed that a “just-right” levels are required for ALL as too much will lead to apoptosis and too-little will lead to suppression of proliferation. However, as shown in Figure 1A, the FPKM of DUX4 ranges from 30 to 500 in patients with IGH-DUX4, which seems not so “just-right” and might conflict with the so-called “Goldilocks principle”.

[Author Response] We thank the reviewer for this comment. In devising our statement that the “Goldilocks” principle is at play in IGH-DUX4 re-arrangement, we drew on the observation of the reported loss of proliferation upon knockdown of DUX4 in Nalm6 by Yasuda et al.¹ [Nat Genet 48, 569-574 (2016)], together with our experiments presented in this manuscript – notably that we observe increased apoptosis upon overexpression of DUX4 in Nalm6. Clearly a specific expression level in between is required for leukemic proliferation, which led us to hypothesize that the Goldilocks principle is at play.

As the reviewer correctly notes, when considering the absolute levels of DUX4 expression in human IGH-DUX4 patients, a range of expression levels was observed (FPKM = **183.7+/-43.5 95% CI**; min 34.3; max 587.1) (Figure 1B). This range of expression is not unexpected given the likely presence of many contributing factors, including nuclear localisation, changes in chromatin structure and histone modifications, non-coding sense and antisense RNA transcription, epigenetic alterations at the DNA level, feedback signaling from expressed alleles, locus contraction and decontraction, and recruitment to heterochromatin [Corcoran AE, Semin Immunol. 2005]². A wide distribution of FPKM values has also been observed in other cancer driver genes. For example, TP53, a well-known tumor suppressor gene disrupted by mutations or structural variations in 95% of pediatric osteosarcomas, exhibits a wide range of FPKM values (0.77-34.46, mean 11.27+/-3.88 95% CI) in the 29 TP53-defective osteosarcomas even after exclusion of the two outlier samples with FPKM of 226.24 and 910.75. Similarly, TAL1, an oncogene known to be activated in a subtype of lineage ALL, also exhibits a wide range of expression in the TAL1-subtype of T-ALL ranging from 0.76 to 48.35 (median 11.29). Both datasets were analyzed in our previous studies (Chen et al, 2015³, Liu et al, 2017⁴) and can be queried in our web portal ProteinPaint (<https://proteinpaint.stjude.org/>).

Importantly, we also observe: 1) in Figure 1B there is also variation of the Igu expression level among these patients (FPKM = **780.4+/-151.6 95% CI**; min 258.7, max 1913.0); 2) among patients, DUX4 expression represents a fraction of the Igu expression level (**26.8%+/-6.4% 95% CI**); and 3) there is a positive correlation (**Pearson's r=0.554, p-value=0.001**) between Igu and DUX4 expression (Figure 1A). These data likely indicate that the observed DUX4 expression variation in patient samples reflects patient-specific mechanics of expression control of Igu. Notably, the case with the lowest DUX4 expression (FPKM ~30 for DUX4, FPKM ~110 for Igu) as well as the one with the highest DUX4 expression (FPKM 500 for DUX4, FPKM ~1500 for Igu) both fit well with this model. In the revision, we have included the following statement after proposing the “Goldilocks model” in Discussion: “*Indeed, among patient samples that harbor IGH-DUX4 fusion, DUX4 expression represents a fraction of the Igu expression level (26.8%+/-6.4% 95% CI) with a positive correlation (Pearson's r=0.554, p-value=0.001, two tailed p value by the Pearson's correlation test) between Igu and DUX4 expression (Figure 1A).*”

2 Allele specific knockdown experiments may provide the direct evidence for the IGH-DUX4 translocation occurs on the silenced IGH allele, although it is certainly difficult to develop suitable sgRNAs that target the individual allele.

[Author Response].

Based on our understanding, the reviewer suggests designing sgRNAs specifically targeting the silenced IGH allele in order to provide direct evidence that the translocation occurred on the silenced IGH allele. Notwithstanding the technical challenge involved in such experiment as pointed by the reviewer, while the result will provide direct evidence that DUX4 was transcribed from the re-arranged allele, it will not yield any further insight regarding the epigenetic status of the rearranged allele. We believe that the full-length Iso-seq RNA generated by PacBio has already provided the definitive answer to the question on IGH-DUX4 translocation, as all transcripts containing DUX4 sequence are chimeric (Supplementary Fig. 1). The epigenetic status of the re-arranged haplotype (assembled by haplotype phasing using digital long-read by chromium WGS) has been resolved through the extensive epigenetic profiling carried out in this study, and the resulting data, including H3K27Ac, ATAC-seq and WGBS, have been presented in Fig. 2B & C of the current version of the manuscript.

In revision, we have included the following statement to reaffirm that DUX4 gene was transcribed only from the re-arranged haplotype in Results in the section of **Allele specificity of IGH-DUX4 translocation** as follows: “all Iso-Seq RNA reads that contain *DUX4* are chimeric and can be mapped to the assembled *IGH-DUX4* haplotype (Supplementary Fig. 1b).”

Reviewer #3 (Remarks to the Author):

minor remarks

some typos?

eg have shown or showed instead of shown?

[Author Response] We thank the reviewer for pointing out this typo and have made the correction.

Reviewer #4 (Remarks to the Author):

The authors have carefully addressed my concerns, implemented important additional controls and toned down their conclusions, resulting in a much improved manuscript.

[Author Response] We thank the reviewer for their help with improving the manuscript.

References

1. Yasuda T, *et al.* Recurrent DUX4 fusions in B cell acute lymphoblastic leukemia of adolescents and young adults. *Nat Genet* **48**, 569-574 (2016).
2. Corcoran AE. Immunoglobulin locus silencing and allelic exclusion. *Semin Immunol* **17**, 141-154 (2005).
3. Chen X, *et al.* Recurrent somatic structural variations contribute to tumorigenesis in pediatric osteosarcoma. *Cell Rep* **7**, 104-112 (2014).
4. Liu Y, *et al.* The genomic landscape of pediatric and young adult T-lineage acute lymphoblastic leukemia. *Nat Genet* **49**, 1211-1218 (2017).